# Optocontrol of glutamate receptor activity by single side-chain photoisomerization

Viktoria Klippenstein[1], Christian Hoppmann[2,3], Shixin Ye[1,4], Lei Wang[2,3], Pierre Paoletti[1]*

[1]Institut de Biologie de l'École Normale Supérieure, Ecole Normale Supérieure, CNRS, INSERM, PSL Research University, Paris, France; [2]Department of Pharmaceutical Chemistry, University of California, San Francisco, San Francisco, United States; [3]Cardiovascular Research Institute, University of California, San Francisco, San Francisco, United States; [4]Laboratory of Computational and Quantitative Biology, Université Pierre-et-Marie-Curie, CNRS, Paris, France

**Abstract** Engineering light-sensitivity into proteins has wide ranging applications in molecular studies and neuroscience. Commonly used tethered photoswitchable ligands, however, require solvent-accessible protein labeling, face structural constrains, and are bulky. Here, we designed a set of optocontrollable NMDA receptors by directly incorporating single photoswitchable amino acids (PSAAs) providing genetic encodability, reversibility, and site tolerance. We identified several positions within the multi-domain receptor endowing robust photomodulation. PSAA photoisomerization at the GluN1 clamshell hinge is sufficient to control glycine sensitivity and activation efficacy. Strikingly, in the pore domain, flipping of a M3 residue within a conserved transmembrane cavity impacts both gating and permeation properties. Our study demonstrates the first detection of molecular rearrangements in real-time due to the reversible light-switching of single amino acid side-chains, adding a dynamic dimension to protein site-directed mutagenesis. This novel approach to interrogate neuronal protein function has general applicability in the fast expanding field of optopharmacology.

*For correspondence: pierre.paoletti@ens.fr

Competing interests: The authors declare that no competing interests exist.

## Introduction

The vast majority of excitatory neurotransmission in the mammalian brain is mediated by ionotropic glutamate receptors (iGluRs), which are classified into AMPA, kainate, and NMDA receptors (*Traynelis et al., 2010*). NMDA receptors (NMDARs) stand out from other iGluRs by their unique capability to induce long-term synaptic plasticity that underlies higher cognitive functions such as learning and memory. They are also targets of therapeutic interest since their dysfunction is associated with numerous neurological and psychiatric disorders such as schizophrenia, mental retardation, and epilepsy (*Paoletti et al., 2013*).

At the structural level, NMDARs are massive (>550 kD) tetrameric complexes, typically composed of two GluN1 and two GluN2 subunits. Representative of their receptor family, NMDARs show a layered organization with an amino-terminal domain (NTD) and an agonist-binding domain (ABD) extruding into the extracellular space, an intracellular carboxy-terminal domain (CTD), and a transmembrane domain (TMD) that contains the selective ion channel pore (*Karakas and Furukawa, 2014*; *Lee et al., 2014*). A main feature of native NMDARs is their broad molecular heterogeneity that translates into a wide variety of receptor subtypes with distinct biophysical, pharmacological, and signaling properties. This is further diversified by their differential location between brain

**eLife digest** Nerve cells communicate with each other by releasing chemicals, also known as neurotransmitters, from one cell to the next. Once released, these neurotransmitters bind to specific docking stations, called receptors, which are located on the surface of the neighboring cell. Due to changes in neurotransmitter release or the receptor number, the connections between neurons can either strengthen or weaken over time. This process, called synaptic plasticity, forms the basis of learning and memory. One of the key players in synaptic plasticity are NMDA receptors, and if these receptors are faulty, it can cause disorders such as schizophrenia or epilepsy.

NMDAs are a large family of receptors that have many receptor subtypes, each with specific properties. Every subtype is composed of four varying subunits. It is still unclear how these different receptor subtypes contribute to synaptic plasticity and new methods are needed to resolve this puzzle.

An emerging strategy to study brain receptors is to engineer them so that they can be controlled with light. One approach to provide light-sensitivity uses molecules that act as 'light switches'. These switches change their shape when exposed to specific colors of light and this way, turn a receptor on or off. However, commonly used light switches are often very large, meaning that they can only be introduced at specific sites in a receptor, and have limited ability to change the shape of a receptor.

Klippenstein et al. have now generated a small light switch molecule with the size of a single amino acid side-chain that, in theory, could replace any of the usual amino acids in the NMDA receptor. Different locations for the light switch were tested to identify those that changed the activity of the receptor. When the receptors were stimulated with light, the light switch changed its shape, which in turn influenced the shape of the receptor. This meant that, depending on which amino acid in the receptor had been replaced with the light switch, light could be used to control the receptor activity in different ways.

This new approach of using integrated light switches allows NMDA receptors to be controlled in a fast and reversible manner using something as simple as a beam of light. Further research will use the toolset of light-controllable receptors to study how the different NMDA receptor subtypes affect synaptic plasticity in the normal and diseased brain.

regions, developmental stages, and even subcellular localizations, supporting the idea that each receptor subpopulation is tailored to match the strict requirements of specific neuronal functions (*Paoletti et al., 2013*). Up to now, putative contributions made by the individual NMDAR subunits to specific neuronal functions are largely unknown or controversial. The GluN2 subunits, of which there are four subtypes (GluN2A-D), are the key determinants of the receptor's functional diversity (*Paoletti, 2011*; *Wyllie et al., 2013*; *Glasgow et al., 2015*). Among those, GluN2A and GluN2B are the major subunits in the adult brain, endowing receptors with distinct charge transfer capacities during activity-dependent synaptic transmission. For instance, the influence of the GluN2A/2B ratio on the polarity of synaptic plasticity – whether long-term potentiation (LTP) or long-term depression (LTD) is induced – is still debated (*Yashiro and Philpot, 2008*; *Paoletti et al., 2013*). Likewise, the role of NMDAR at synaptic and (GluN2B-enriched) extrasynaptic sites in cell survival remains largely contentious (*Parsons and Raymond, 2014*). Thus, the modality of GluN2-subtype-driven fine-tuning of information processing in distinct brain regions, within individual neurons, and at individual excitatory synapses remains to be resolved.

By providing precise ways to manipulate endogenous signaling proteins, optopharmacological approaches open new avenues for deciphering the molecular basis of physiological function. Optopharmacology combines the power of optics, endowing high spatiotemporal resolution, with that of genetics and pharmacology, to achieve unique photocontrol on the molecular receptor level (*Kramer et al., 2013*). One attractive strategy to engineer light-responsiveness relies on the introduction of azobenzene moieties. Upon exposures to light of different wavelengths, azobenzene groups undergo a reversible photoswitching between two different configurations – a compact *cis*- and a stretched *trans*-configuration (*Beharry and Woolley, 2011*; *Broichhagen and Trauner, 2014*).

The use of azobenzene offers high quantum yield, minimal photo-bleaching, excellent photo-stability, and has the major benefit of functional reversibility. In the last decade, chemical tethering of photoswitchable azobenzene-coupled ligands has been shown to be an elegant and powerful method to engineer light-responsiveness in neuronal receptors. This strategy has allowed to photo-agonize or -antagonize different classes of glutamate receptors (*Volgraf et al., 2006*; *Reiner et al., 2015*; *Berlin et al., 2016*) and diverse members of other neurotransmitter receptor families (*Tochitsky et al., 2012*; *Lemoine et al., 2013*; *Browne et al., 2014*; *Lin et al., 2015*). This approach, however, requires proper protein conjugation with the azobenzene-coupled ligand and is restricted to pharmacologically-characterized sites that are accessible to the external solvent, excluding intracellular and transmembrane protein domains.

The incorporation of photoreactive groups into the receptor itself, using unnatural amino acids (UAAs), provides an alternative to achieve optical control over receptor activity. The UAA methodology relies on the re-assignment of a stop codon (usually the Amber stop codon) by a suppressor tRNA aminoacylated with the desired UAA (*Wang et al., 2001*; *Chin, 2014*; *Leisle et al., 2015*; *Liu and Schultz, 2010*). Orthogonal tRNA/synthetase pairs enable efficient and direct incorporation of genetically-encoded UAAs into receptors expressed in simple cell lines, neuronal cultures, brain slices, and even whole organisms (*Wang et al., 2007*; *Kang et al., 2013*; *Klippenstein et al., 2014*; *Zhu et al., 2014*; *Ernst et al., 2016*). Important in the context of our study, light-triggered inactivation of AMPA and NMDA receptors was recently demonstrated following incorporation of the photocrosslinking UAAs Azido-phenylalanine (AzF) and Benzoyl-phenylalanine (BzF) at specific interface sites (*Klippenstein et al., 2014*; *Zhu et al., 2014*; *Tian and Ye, 2016*). Another type of UAAs, decorated with a photocage, was shown to provide control over neuronal excitability following incorporation into $K^+$-channels (*Kang et al., 2013*). Photocrosslinking or photocaged UAAs offer excellent site tolerance and molecular specificity (even at solvent-inaccessible sites), however, they require prolonged exposures to light (time scale of several seconds or more), have a possibly fragile photostability (in case of AzF), and crucially, lack reversibility.

In this study, we combined the advantages of UAA incorporation and azobenzene photochemistry by implementing azobenzene-based photoswitchable UAAs (PSAAs) (*Bose et al., 2006*; *Hoppmann et al., 2014*) in order to achieve fast and reversible photocontrol over a set of NMDAR subunits. We demonstrate that genetically encodable PSAAs provide an efficient and flexible approach to install precise and reversible light-sensitivity on a key neuronal receptor. By revealing novel structural determinants involved in controlling NMDAR function, we further provide evidence that PSAAs are powerful tools to probe receptor biophysics. In particular, our approach allows targeting transmembrane pore sites whose conformational changes during receptor activity remain poorly understood. The development of photoswitchable NMDARs with genetic encodability, as presented here, should be valuable to manipulate NMDAR signaling and unmask GluN2-specific roles in neuronal function. We expect these principles to be generally applicable to other brain proteins, enabling optical investigation of a range of receptors and ion channels both in recombinant and native systems. Differences and advantages of the PSAA approach compared to classical mutagenesis are discussed as well as the technical requirements for its in vivo application.

## Results

### Genetically encoding PSAAs into NMDAR subunits

Recently, we and others have successfully exploited the genetic code expansion methodology to design light-sensitive iGluRs through incorporation of photocrosslinking UAAs (*Klippenstein et al., 2014*; *Zhu et al., 2014*; *Tian and Ye, 2016*). Here, following similar principles, we expanded the approach to generate NMDARs containing azobenzene-based photoswitchable UAAs, which have the major advantage of reversibility (*Figure 1*). To test the feasibility of genetically-encoded PSAAs in NMDARs, we co-expressed specific NMDAR subunits containing an introduced Amber stop codon with a genetically engineered orthogonal tRNA/aminoacyl synthetase pair derived from the methanogenic archea *Methanosarcina mazei* (*Mm*) in HEK cells (*Figure 1a*). This tRNA/synthetase pair, evolved from the pyrrolysine tRNA/synthetase pair (tRNA$^{Pyl}$–*Mm*PSCAA-RS), allows incorporation of azobenzene-based PSAAs into *E. coli* and mammalian cells in a site-specific manner (*Hoppmann et al., 2014*). The simple photoisomerization from the planar (*trans*) to the bent (*cis*)

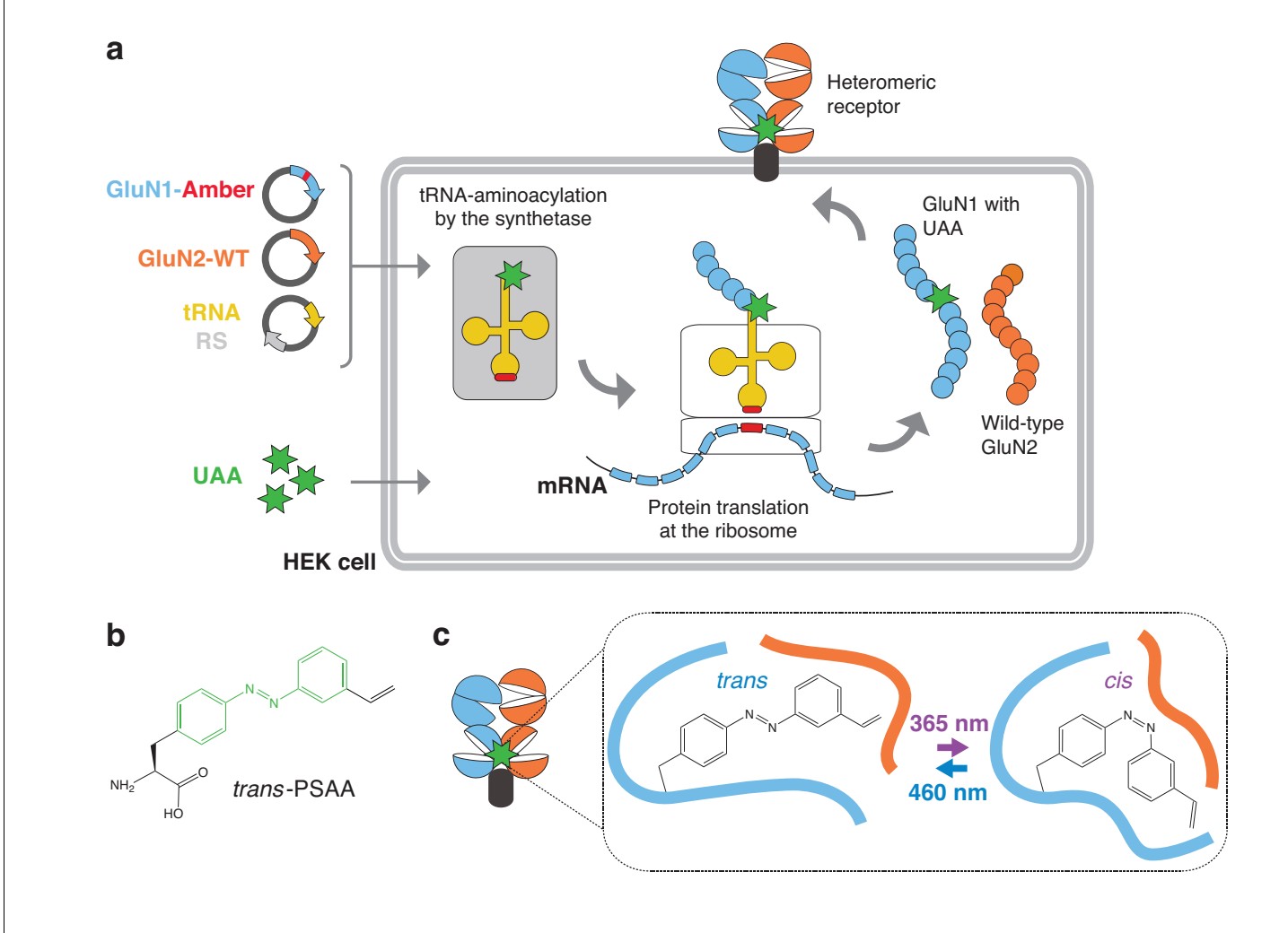

**Figure 1.** General principle for genetic encoding photoswitchable UAAs into membrane receptors. (**a**) Schematic representation of the UAA methodology applied to NMDARs. A gene encoding a membrane protein of interest (here shown for the NMDAR GluN1 subunit; *blue*) and containing an introduced Amber stop codon (TAG, *red*) at a desired position is co-transfected into HEK cells with vectors encoding a WT GluN2 subunit and an orthogonal suppressor tRNA (*yellow*) / aminoacyl synthetase (RS; *grey*) pair. The cells are incubated in the presence of the photoswitchable UAA (PSAA, *green asterisks*) in the culture medium. Within the cell, the orthogonal RS specifically aminoacylates the suppressor tRNA with the PSAA. At the ribosome level, the PSAA is incorporated in response to the Amber codon on the mRNA by the complementary anticodon CUA on the suppressor tRNA (*red*). After release from the ribosome, the full-length GluN1 polypeptide chain, site-specifically carrying the PSAA, assembles into a functioning receptor expressed at the cell surface. (**b**) Structure of the PSAA. The photoisomerizable azobenzene unit is highlighted in *green*. Additionally, a functional alkene group, which allows a covalent click reaction with nearby cysteines, is attached to the azobenzene unit. (**c**) Diagram of a NMDAR GluN1/GluN2 heterodimer carrying a PSAA at the ABD dimer interface. Toggling between the *trans*- and *cis*-configuration is induced by light exposure at 460 or 365 nm, respectively. This simple change in side chain geometry, when placed at a receptor key moving site, allows inducing remarkable structural changes that can impact receptor activity.

configuration of the azobenzene moiety results in a pronounced change in shape, with a reduction in the end-to-end distance between the two rings of 3.5 Å (*Beharry and Woolley, 2011*) (*Figure 1b and c*). The Alkene moiety of PSAA provides an additional functionality for click reactions and formation of photobridges with nearby cysteines (*Hoppmann et al., 2014*, *2011*). Here, we reasoned that the single side-chain flip of the azobenzene moiety may be sufficient to drive structural changes and impact receptor functionality when introduced at specific subunit locations (*Figure 1c*). Thus, we introduced Amber mutations, one at a time, at various sites within the GluN1 or GluN2 subunit and assessed receptor functionality using electrophysiological patch-clamp recordings (*Table 1*). Three

**Table 1.** Screening for expression and photosensitivity of NMDAR Amber mutants. Overview of all Amber positions screened for PSAA insertion within the NTD, ABD, and TMD. PSAA was introduced to either the GluN1, GluN2A, or GluN2B subunit. A total of 27 different combinations were tested. Among those, 13 gave co-agonist evoked currents (*green boxes*), the rest did not result in functional receptors (*orange boxes*). Eight subunit combinations assembled into functional receptors, whose activity could be reversibly modulated by light (↓ UV-triggered photoinactivation; ↑ UV-triggered photopotentiation).

| | GluN1 | GluN2A | GluN2B | Expression | Photomodulation |
|---|---|---|---|---|---|
| **WT** | wt | wt | | (green) | (orange) |
| | wt | | wt | (green) | (orange) |
| **NTD** | Y109 | wt | | (green) | ↓ |
| | Y109 | | wt | (green) | ↓ |
| | wt | Y281 | | (green) | (orange) |
| | wt | | Y282 | (green) | (orange) |
| **ABD** | P521 | wt | | (green) | (orange) |
| | P532 | wt | | (green) | ↓ |
| | P532 | | wt | (orange) | |
| | Y535 | wt | | (green) | ↓ |
| | wt | P527 | | (orange) | |
| | wt | E530 | | (orange) | |
| | wt | | P528 | (orange) | |
| | wt | | E531 | (orange) | |
| | E698 | wt | | (green) | (orange) |
| **TMD** | F554 | wt | | (green) | ↑ |
| | W563 | wt | | (green) | ↑ |
| | W578 | wt | | (orange) | |
| | W608 | wt | | (orange) | |
| | W611 | wt | | (green) | (orange) |
| | wt | S632 | | (orange) | |
| | W636 | wt | | (orange) | |
| | wt | Y645 | | (orange) | |
| | wt | | Y646 | (orange) | |
| | Y647 | wt | | (green) | ↑ |
| | Y647 | | wt | (orange) | |
| | F654 | wt | | (green) | ↑ |
| | F654 | | wt | (orange) | |
| | F817 | wt | | (orange) | |

phenotypes were observed in our mutational analysis: (*i*) absence of measurable NMDAR-mediated currents; (*ii*) presence of currents, but lack of photosensitivity; (*iii*) presence of photosensitive currents. We interpreted the first scenario as indicative of improper receptor expression and focused on the latter, demonstrating successful PSAA incorporation into mature, cell surface expressed NMDARs. Promisingly, within each domain of the receptor (NTD, ABD, and TMD), at least one position was identified that endowed significant light-sensitivity (*Table 1*), highlighting the efficacy of the approach.

## Photosensitive ABD domains to control receptor activity with light

The activation of NMDARs requires binding of two different agonists, glutamate and glycine (or D-serine), a unique feature among ligand-gated ion channels. These co-agonists bind at appropriate clamshell-like domains that close thereupon to transduce the signal to the ion channel pore. Within a tetrameric GluN1/GluN2 receptor, the ABDs assemble as two heterodimers with the glycine-binding GluN1 ABD and the glutamate-binding GluN2 ABD, pairing through back-to-back interactions within each dimer (*Furukawa et al., 2005*; *Karakas and Furukawa, 2014*; *Lee et al., 2014*) (*Figure 2a*). To endow NMDARs with optical sensitivity, we initially targeted the GluN1/GluN2A ABD heterodimer interface for PSAA incorporation since this region critically controls receptor functionality (*Furukawa et al., 2005*; *Gielen et al., 2008*; *Borschel et al., 2011*). Replacing the highly conserved proline residue GluN1-P532 by the PSAA resulted in functional receptors, as represented by large and robust co-agonist activated currents (*Figure 2—figure supplement 1*) as well as a pronounced modulation of receptor activity induced by application of light (*Figure 2*). Specifically, following receptor activation with saturating co-agonist concentrations, illumination with UV light (365 nm) to switch the PSAA from the *trans-* to the *cis-*isomer, produced a remarkable current inhibition of 48.9 ± 1% ($n$ = 23; *Figure 2b and g*). Crucially, this effect could be fully reversed by blue light (460 nm) that regenerates the *trans-*version of the PSAA. The degree of photoinactivation was similar when UV illumination occurred prior to agonist application (48.4 ± 1.6% [$n$ = 7]; *Figure 2c and d*) or during application of competitive antagonists (*Figure 2—figure supplement 2*), indicating that the effect is not dependent on the functional state of the receptor. Application of blue light in the resting state was ineffective to alter the current amplitude (*Figure 2c and d*), in agreement with the PSAA azobenzene moiety adopting essentially 100% of the *trans-*isomer in the dark (*Beharry and Woolley, 2011*). Importantly, both wavelengths of light had virtually no effect on wild-type (WT) GluN1/GluN2A receptors (*Figure 2e–g*). To confirm specific incorporation of PSAA, we performed identical transfections of mutant NMDAR subunits and the tRNA/synthetase pair, but omitted to supplement the culture medium with the PSAA. In such conditions, the majority of transfected cells yielded no or tiny NMDAR-mediated currents (few tens of pA at most) that were completely insensitive to UV or blue light, indicating that the Amber codon suppression by endogenous amino acids is negligible (*Figure 2—figure supplement 3a and b*). In PSAA-containing receptors, the degree of UV-induced current inhibition neither depended on the initial level of the peak current amplitude nor on the extent of receptor desensitization (*Figure 2—figure supplement 3c and d*). Overall, these results reveal a high yield of PSAA incorporation at the GluN1-P532 site and strongly support the presence of a single homogenous (PSAA-containing) receptor population at the cell surface. Further evidence for specific PSAA introduction was obtained by exposing P532PSAA mutant NMDARs to the positive allosteric modulator (PAM) GNE-6901. This compound binds the GluN1-GluN2 ABD upper lobe dimer interface (*Hackos et al., 2016*), adjacent to the PSAA incorporation site. Whereas WT receptors were strongly potentiated upon PAM application, GluN1-P532PSAA mutant receptors were entirely unaffected by PAM, but retained light sensitivity in the presence of PAM (*Figure 2—figure supplement 4*), indicating the disruption of the PAM-binding site by PSAA. We found that PSAA-substitution at the ABD site GluN1-Y535 also resulted in a significant UV-induced photoinactivation. The extent of photomodulation was, however, less pronounced compared to GluN1-P532. On that account and due to the lower expression levels of the GluN1-Y535PSAA mutant receptors (*Figure 2—figure supplement 1*), we chose the more robust GluN1-P532 site to examine the photomodulation properties in further detail.

## Photomodulation properties of GluN1-P532PSAA/GluN2A receptors

We first quantified the light responsiveness of the receptor by measuring the current inhibition and re-activation kinetics induced by the wavelength-dependent photoisomerization of the azobenzene moiety. Forward and backward transitions between the fully active and the UV-inactivated state were both satisfactorily fitted by monoexponetial functions with decay time constants in the tens to hundreds of ms range ($\tau_{uv}$ = 500 ± 30 ms [$n$ = 23]; $\tau_{BLUE}$ = 45 ± 1 ms [$n$ = 23]; *Figure 3a*). Additionally, we examined the re-activation kinetics when stepping back to green light (520 nm) that revealed a much slower switch rate ($\tau_{GREEN}$ = 1350 ± 140 ms [$n$ = 5], *Figure 3a*), associated with partial current recovery (76 ± 7% [$n$ = 5], compared to full recovery with blue light). This partial recovery, accompanied by its slowness, is indicative of a mixture of PSAA *cis-* and *trans-* forms at 520 nm and

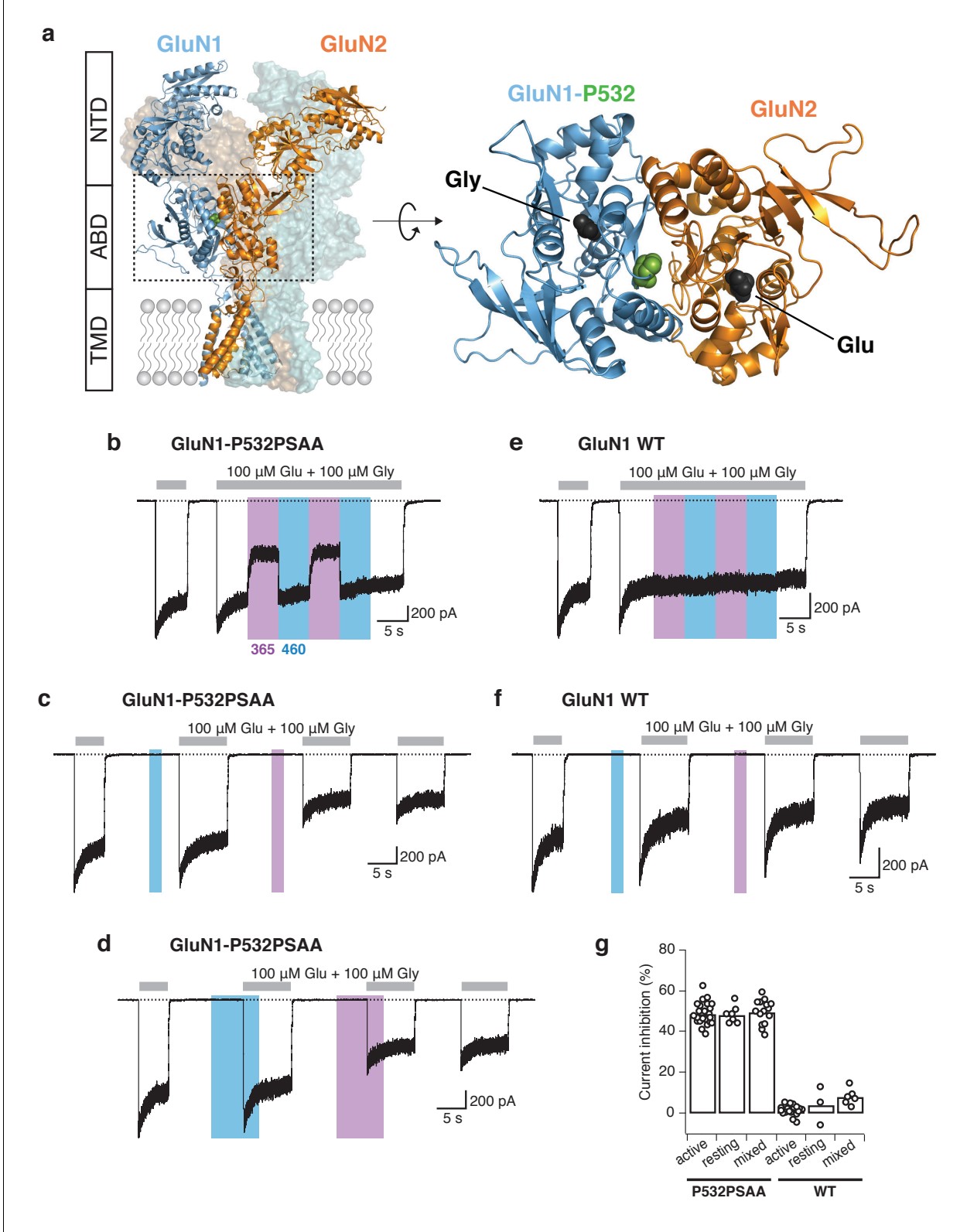

**Figure 2.** Reversible photomodulation of NMDARs with PSAA at the GluN1-P532 site. (a) Structure of a heteromeric GluN1/GluN2B receptor (*left panel*). The two GluN1 subunits are shown in *blue*, the two GluN2 subunits in *orange*. One receptor dimer is highlighted by cartoon representation. The typical layered organization contains the N-terminal domains (NTDs), the agonist-binding domains (ABDs), and the transmembrane domain (TMD). The C-terminal domain (CTD) is absent in the structure. The Amber mutation site GluN1-P532 at the GluN1/GluN2 ABD heterodimer interface is shown as

*Figure 2 continued on next page*

*Figure 2 continued*

*green spheres.* The top view of one ABD dimer highlights the location of the PSAA insertion (*right panel*). (**b**) Representative current trace of GluN1-P532PSAA/GluN2A receptors showing reversible photomodulation following 5 s of UV and blue light during the application of saturating co-agonists. In this example, the UV-driven current reduction was 46%. (**c**) UV (2 s) in the resting state on the same cell as in *b* gave a current inhibition of 51%. (**d**) The photoinactivation degree was similar when UV was applied in the 'mixed' state (50%). (**e**) As in *b*, for WT GluN1/GluN2A receptors. Nearly no modulation of the current amplitude was observed (2% inhibition with UV). (**f**) As in *c*, for WT receptors (13% inhibition with UV). (**g**) Summary of UV inhibition degrees for the PSAA-mutant and WT receptors following UV exposures in different states. For the PSAA-mutant, mean photoinactivation values are (in %): $Inh_{active}$ = 48.9 ± 1 ($n$ = 23), $Inh_{rest}$ = 48.4 ± 2 ($n$ = 7), $Inh_{mixed}$ = 49.7 ± 2 ($n$ = 14). For WT receptors: $Inh_{active}$ = 1.3 ± 0.6 ($n$ = 18), $Inh_{rest}$ = 4 ± 5 ($n$ = 3), $Inh_{mixed}$ = 7.9 ± 2 ($n$ = 6).

The following figure supplements are available for figure 2:

**Figure supplement 1.** : Summary of expression levels for wild-type receptors and all GluN1 Amber mutant receptors that exhibited photomodulation following introduction of PSAA.

**Figure supplement 2.** The presence of the competitive antagonists APV and DCKA does not impact the UV-mediated current inhibition for GluN1-P532PSAA/GluN2A mutant receptors.

**Figure supplement 3.** High efficiency of PSAA-incorporation at the GluN1-P532Amber site.

**Figure supplement 4.** PSAA incorporation at the GluN1-P532 Amber site disrupts PAM action.

is consistent with the green light providing less energy and being less efficient than blue light to reverse PSAA from the *cis*- to the *trans*-isomer (*Figure 3—figure supplement 1*) (*Beharry and Woolley, 2011*; *Hoppmann et al., 2014*, *2011*). We further studied the dependence of the UV-induced photoinactivation on the duration and power of the UV irradiation (*Figure 3—figure supplement 2*). Both the extent and time course of photoinactivation were measured. Since isomerization events of free azobenzene groups are known to occur on a picosecond timescale (*Beharry and Woolley, 2011*), substantially faster than most biological processes, we tested if short light pulses (<5 s, the duration used so far) were sufficient to trigger equal photoresponses. By varying the light duration between 100 ms and 5 s, we identified a UV interval length of 1 s to be the minimum for maximal current inactivation (*Figure 3—figure supplement 2a–d*). Similar experiments including variations of UV light intensities (10–100%) revealed a strong impact on the photoinactivation kinetics – with lower intensities resulting in slower kinetics – but much less on the degree of inhibition (*Figure 3—figure supplement 2e–h*). Importantly, however, there was no difference in the photomodulation properties (time-course, degree of current inhibition) between a UV intensity of 75% and full LED power indicating that the conditions used in most experiments (1–5 s of UV at full power) enabled full photoregulation. Strikingly, although light-induced changes in receptor activity were all relatively fast on a biological time scale, they were surprisingly slow in regard to the (ultrafast) chemistry of azobenzene photoisomerization (see above). This difference could stem from protein conformation changes needed to allow isomerization to local interactions with nearby residues of the PSAA embedded into the protein.

We next assessed the robustness and stability of the photoresponses. Once initiated by a short pulse of UV, the photoinactivation remained stable for extended periods of time (minutes), well beyond the presence of UV light and irrespective of the presence or absence of agonists (*Figure 3b and c*). Thus, the slow spontaneous thermal *cis*-to-*trans* isomerization of azobenzene as seen in solution (*Beharry and Woolley, 2011*; *Hoppmann et al., 2011*) is preserved in the context of the protein-embedded PSAA. The photoinactivated state was also highly stable during elongated exposures to UV (30 s; *Figure 3d*). Following such a prolonged photoinactivation, immediate resetting to full receptor activity could still be achieved by a brief pulse of blue light, demonstrating minimal photobleaching of the azobenzene moiety and lack of phototoxicity. Again, similar light protocols performed on WT receptors showed no photosensitivity (*Figure 3—figure supplement 3*). Finally, repetitive cycles of illumination showed that the photoresponses could sustain multiple rounds of *cis*-to-*trans* photoisomerization with no detectable fatigability (*Figure 3e*). In summary, all

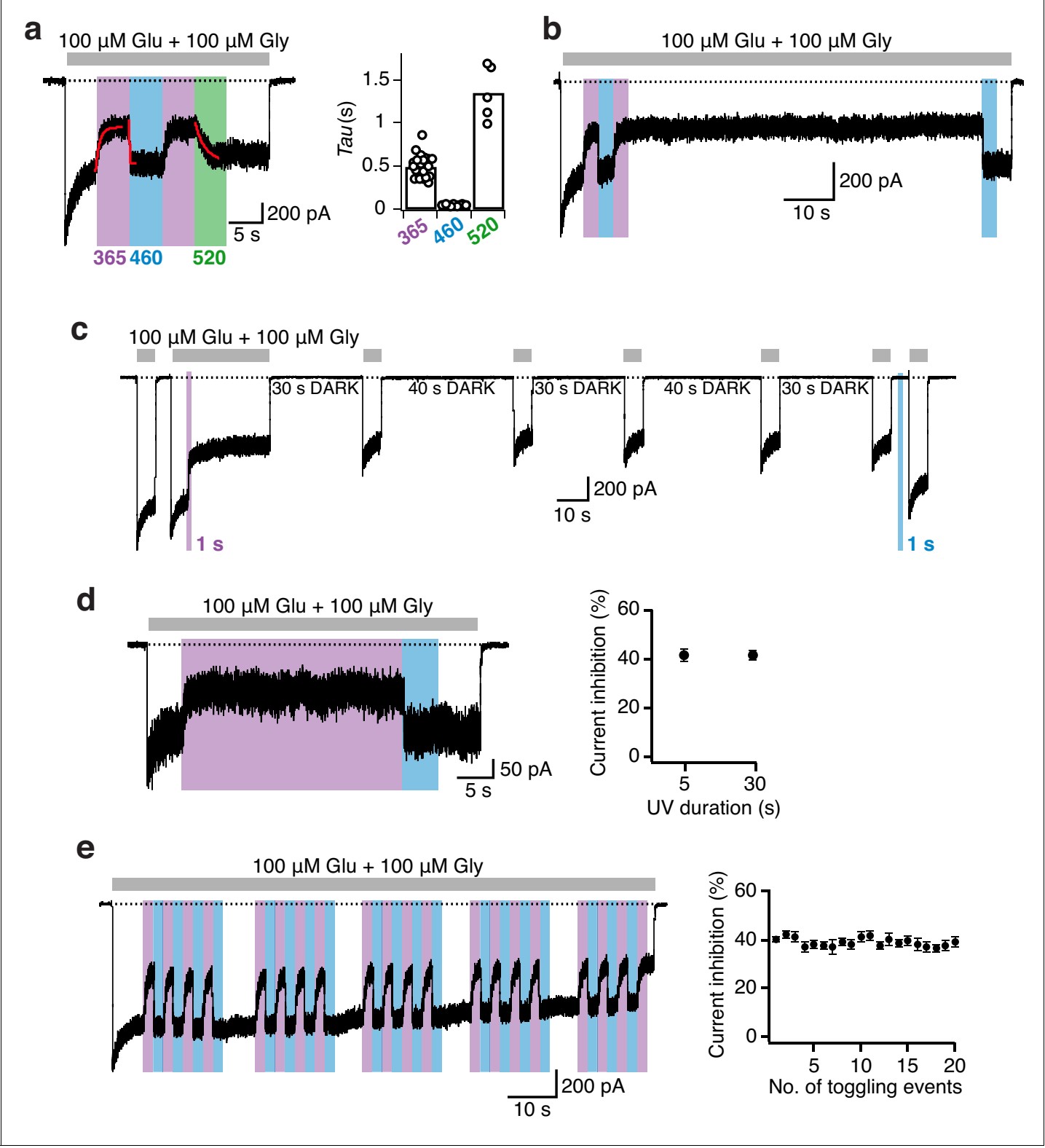

**Figure 3.** Photomodulation properties of GluN1-P532PSAA/GluN2A receptors. (a) Kinetics of photoinhibition and its recovery. Photoinactivation following UV (5 s) was slower (τ = 520 ms) compared to the re-activation induced by blue light (τ = 35 ms), as obtained by monoexponential fits. Much slower re-activation kinetics (τ = 1300 ms) and an intermediate state of recovery are observed upon green light exposure (*left panel*). The average exponential time constants for the different wavelengths of light (5 s each) are (in ms): $\tau_{365nm}$ = 500 ± 30 (*n* = 23), $\tau_{460nm}$ = 45 ± 1 (*n* = 23), $\tau_{520nm}$ = 1350 ± 140 (*n* = 5) (*right panel*). (b) Representative current trace demonstrating the persistence of photoinhibition following a brief (3 s) UV

*Figure 3 continued on next page*

*Figure 3 continued*

illumination. (**c**) The *cis*-isoform, induced by UV (1 s), inhibited 40% of the current and was stable over minutes. Further co-agonist activation, interrupted by periods in the dark, did not affect the degree of the photoinhibition. A short pulse of blue light (1 s) allowed full recovery to the initial current amplitude. (**d**) Stability of the photoresponse during prolonged UV exposures (30 s), as shown in the example trace (*left*). The mean current inhibition degrees (in %) following 5 and 30 s of UV are: $Inh_{5s} = 42 \pm 3$, $Inh_{30s} = 41 \pm 2$ ($n = 3$; *right panel*). (**e**) Example trace demonstrating the stability of the photoresponse over 20 PSAA *cis-trans* toggling events. The degree of receptor modulation remained stable throughout the recording (*left panel*). On average, the first UV pulse gave a current reduction of $40 \pm 1\%$. The averaged extent of UV inhibition following 20 toggling events was $39 \pm 2\%$ ($n = 6$; *right*).

The following figure supplements are available for figure 3:

**Figure supplement 1.** : The UV/Vis spectra of the different PSAA photoisomers.

**Figure supplement 2.** Impact of UV duration and intensity on the photomodulation properties of GluN1-P532PSAA/GluN2A receptors.

**Figure supplement 3.** Absence of photoresponses in GluN1/GluN2A WT receptors.

---

observed photoresponses were fully reversible, thermally bi-stable, and displayed reproducible kinetics in a millisecond-to-second time range.

## Optical manipulation of receptor activity and co-agonist sensitivity

The GluN1/GluN2 ABD dimer interface is a central structural determinant of NMDAR activity, which impinges on several key receptor properties, including agonist sensitivity, deactivation kinetics, as well as channel open probability (*Furukawa et al., 2005*; *Gielen et al., 2008*; *Borschel et al., 2011*). We therefore determined the influence of GluN1-P532PSAA photoswitching on NMDAR gating parameters. To examine the channel maximal open probability ($P_o$), we measured the kinetics of current inhibition by the open channel blocker MK-801, a method classically used to index receptor channel $P_o$ (*Zhu et al., 2014*). As indicated by the modestly slower MK-801 blocking rate, the basal receptor $P_o$ was slightly reduced by introducing the PSAA at the GluN1-P532 site per se (dark state; ~1.3 fold slower kinetics; p=0.27). Critically, under UV exposure, MK-801 inhibition kinetics occurred at ~2 fold slower rates, indicating a pronounced decrease in $P_o$ in the photoinactivated state of PSAA-mutant channels (*Figure 4a* and *Figure 4—figure supplement 1*). In contrast, no change in $P_o$ was observed for WT receptors upon exposure to UV or blue light (*Figure 4—figure supplement 1*). Thus, the PSAA introduction and modulation by light at position GluN1-P532 confers direct control of receptor channel $P_o$.

To explore the inter-relationship between agonist occupancy and photomodulation, we next generated full agonist dose-response profiles in the dark or under UV or blue light. Introducing the UAA per se induced a similar reduction of both glutamate and glycine sensitivity, compared to WT receptors (~3 fold increase in $EC_{50}$; *Figure 4b and c*). Upon illumination, however, a striking difference between the two co-agonists arose. The sensitivity to glutamate was found to be equal in all light conditions tested (dark, UV, and blue). In contrast, the sensitivity to glycine was profoundly affected by the change in wavelength. Specifically, inducing the *cis*-form by UV strongly decreased the sensitivity to glycine ($EC_{50}$ increased by ~20 fold compared to WT receptors), an effect that could be subsequently reversed with blue illumination. In accordance with these light effects on glycine potency, the extent of UV-induced current inhibition was highly dependent on glycine concentrations, displaying an inverse correlation (the lower the concentration the larger the photoinhibition; *Figure 4d*). Remarkably, at 1 µM glycine, the reduction in current amplitude was extensive, approaching nearly complete receptor silencing ($94.3 \pm 1\%$ inhibition [$n = 3$]). Finally, to gain further insights into the effects of azobenzene photoswitching on the agonist-receptor interaction, we measured glutamate and glycine deactivation kinetics following short (800 ms) jumps into saturating agonist concentrations under different light exposures. Glycine deactivation kinetics was significantly fastened when switching from dark to UV, in good agreement with the decreased glycine sensitivity observed at equilibrium (see *Figure 4b*). This effect was fully reversed with blue light (*Figure 4e*). For glutamate, the deactivation kinetics were indistinguishable between dark, UV, and blue states (*Figure 4f*), as expected from the insensitivity of the glutamate dose-response curves to light (see *Figure 4c*).

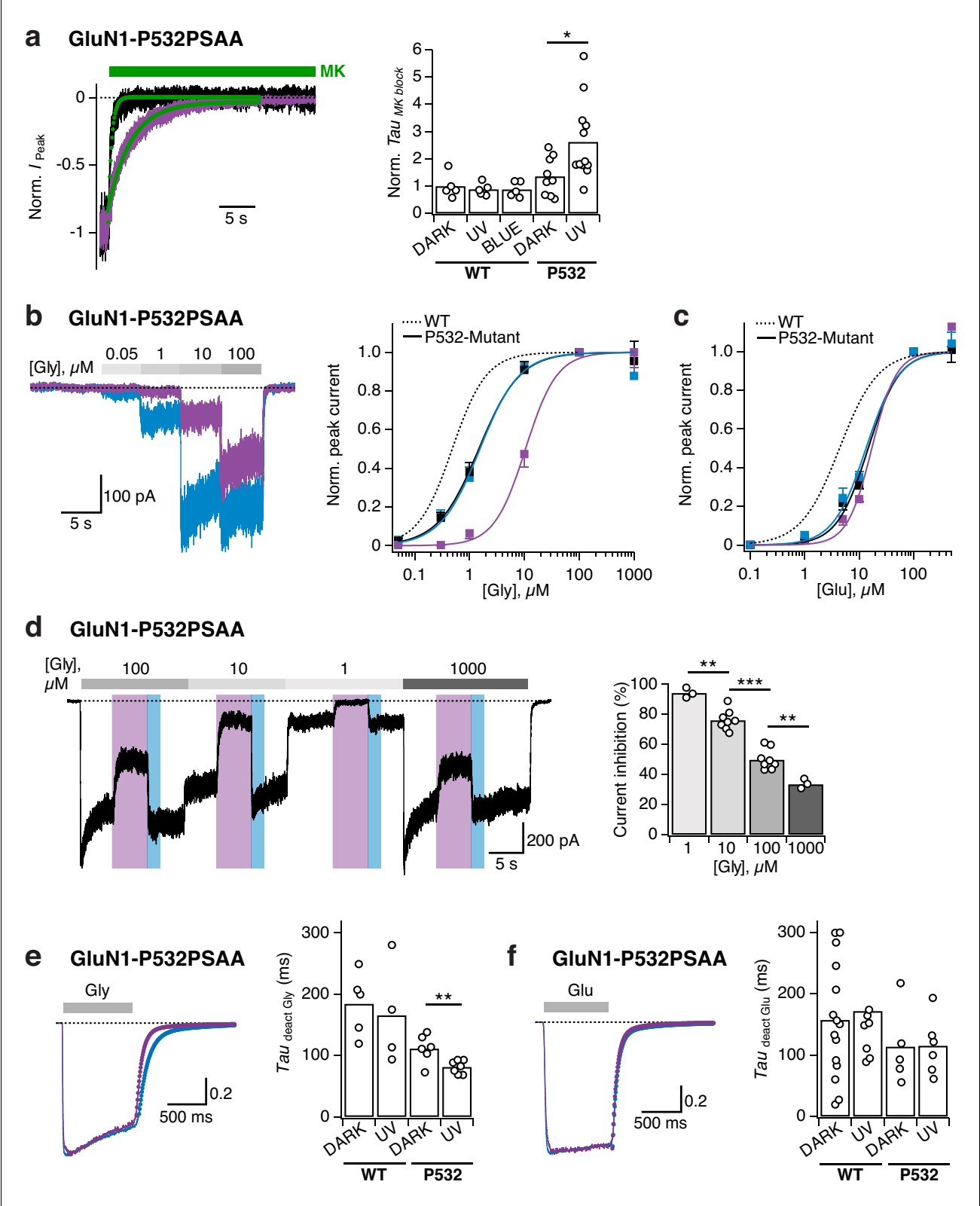

**Figure 4.** Reversible photomodulation of channel activity and glycine sensitivity of GluN1-P532PSAA/GluN2A receptors. (**a**) Normalized example current traces demonstrating the distinct MK-801 inhibition kinetics in the dark and UV-state ($\tau_{DARK}$ = 470 ms; $\tau_{UV}$ = 3200 ms; *left panel*). Pooled values (*right panel*) under various light conditions for WT and PSAA-mutant receptors (*p=0.016). (**b**) Glycine dose-response traces under UV or blue illumination (*left panel*). Glycine DRCs (*right panel*) for WT (*dashed line*) and PSAA-mutant (*full lines*) in the dark (*black line*) or under UV or blue light. $EC_{50}$ values are (in

*Figure 4 continued*

μM): $EC_{50,DARK}$ = 0.5 ($n$ = 5 for each point) for WT and $EC_{50,DARK}$ = 1.5 ($n$ = 3–12), $EC_{50,BLUE}$ = 1.5 ($n$ = 3–8), $EC_{50,UV}$ = 10.5 ($n$ = 3–7) for PSAA-mutant receptors. (c) As in $b$, sensitivity to glutamate. $EC_{50}$ values are: $EC_{50,DARK}$ = 4.4 ($n$ = 8–9) for WT and $EC_{50,DARK}$ = 14.6 ($n$ = 4–11), $EC_{50,BLUE}$ = 13.3 ($n$ = 3–8), $EC_{50,UV}$ = 17.1 ($n$ = 3–8) for PSAA-mutants. (d) Representative current trace demonstrating the photomodulation properties under various glycine concentrations (1, 10, 100 or 1000 μM). The mean photoinhibition degrees are (in %, *right panel*): $Inh_1$ = 94.3 ± 2 ($n$ = 3), $Inh_{10}$ = 76.1 ± 2 ($n$ = 8), $Inh_{100}$ = 49.9 ± 2.5 ($n$ = 8), $Inh_{1000}$ = 33.5 ± 2 ($n$ = 3; ***p<0.001; **p<0.01). (e) Glycine deactivation kinetics. Example trace showing glycine dissociation under UV or blue light ($\tau_{deact,UV}$ = 75.4 ms, $\tau_{deact,BLUE}$ = 131.3 ms; *left panel*). The mean weighted deactivation time constants are (in ms; *right panel*): $\tau_{deact,DARK}$ = 184 ± 16 ($n$ = 5) and $\tau_{deact,UV}$ = 166 ± 30 ($n$ = 4; p=0.69) for WT, and $\tau_{deact,DARK}$ = 111 ± 9 ($n$ = 6) and $\tau_{deact,UV}$ = 81 ± 4% ($n$ = 7; **p<0.01) for PSAA-mutants. (f) As in $e$, for glutamate ($\tau_{deact,UV}$ = 77.7 ms, $\tau_{deact,BLUE}$ = 79.2 ms). The deactivation time constants are (in ms): $\tau_{deact,DARK}$ = 157 ± 25 ($n$ = 15) and $\tau_{deact,UV}$ = 172 ± 30 ($n$ = 8; p=0.92) for WT and $\tau_{deact,DARK}$ = 113 ± 36 ($n$ = 5) and $\tau_{deact,UV}$ = 115 ± 21% ($n$ = 6; p=0.97) for PSAA-mutants.

The following figure supplement is available for figure 4:

**Figure supplement 1.** MK-801 inhibition kinetics for GluN1-P532PSAA/GluN2A mutant receptors.

Overall, these results show that photoswitching the azobenzene side chain at GluN1-P532 to the *cis*-state inhibits receptor activity through a dual effect, a drop in gating efficacy (decrease in channel $P_o$) and a reduction in co-agonist affinity (increase in glycine $EC_{50}$). Also, these findings demonstrate the power of PSAAs to optically manipulate key receptor gating properties in a precise and reversible manner.

## Subunit-specific receptor control

We next targeted the NTD region for PSAA introduction. This region, which lies the most distal to the channel pore, forms a major regulatory region that critically influences NMDAR gating and pharmacology in a subunit-specific manner (*Paoletti, 2011*; *Gielen et al., 2009*; *Yuan et al., 2009*). In particular, we previously showed that introduction of the irreversible photocrosslinker AzF at the GluN1-Y109 site at the NTD upper lobe dimer interface allows to irreversibly photoinhibit GluN1/GluN2B, but not GluN1/GluN2A receptors (*Zhu et al., 2014*). Building on this, we turned back to our GluN1-Y109 Amber cDNA mutant and co-expressed it with either the GluN2A or GluN2B subunit and in the presence of the PSAA. Hereby, UV exposure resulted in inhibition of activity of both types of receptors, an effect that was reversed by blue light illumination (*Figure 5*). Importantly, however, the photoinhibition profiles were markedly distinct between the two mutants, with a modest reduction in current amplitude for mutant GluN2A receptors (21 ± 3% [$n$ = 6]), and a profound inhibition for mutant GluN2B receptors (79 ± 2% [$n$ = 10]; *Figure 5*). Therefore, rearrangements at the GluN1-GluN2B NTD dimer interface can directly modulate the level of receptor activity, in line with recent structural studies (*Tajima et al., 2016*). Also, subtle conformational perturbations at this distal interface, triggered by minimal shape modification of a single azobenzene side chain, are sufficient to alter the downstream gating machinery. Finally, these results point to important structural and functional differences in the NTD region between GluN2A and GluN2B receptors (*Zhu et al., 2014*; *Tian and Ye, 2016*; *Romero-Hernandez et al., 2016*), the two main NMDAR subtypes in the adult brain.

## Optical control of gating and permeation properties with PSAA within the TMD

With the major advantage of site tolerance using UAAs, we next placed our photoswitchable probe to various prominent sites within the TMD (*Table 1*). Primarily, we chose aromatic positions to maintain steric properties within the ion channel region and concentrated on the upper part of the TMD, where the channel gate resides (*Chang and Kuo, 2008*; *Karakas and Furukawa, 2014*; *Lee et al., 2014*). We identified a cluster of bulky hydrophobic residues sitting within a TMD cavity at the back of the M3 bundle crossing, to be tolerant to PSAA introduction and to be highly responsive to light exposure (*Figure 6a*). This cluster includes four GluN1 residues – F554 (in the pre-M1 loop), W563 (in the M1 helix), and Y647 and F654 (in the M3 helix) – most of which are highly conserved within the iGluR family (*Alsaloum et al., 2016*) (*Figure 6—figure supplement 1*). Exposure to UV light with the PSAA placed at one of these sites invariably produced a potentiation of receptor activity, with a site-dependent degree of current amplitude increase. This effect could be fully and rapidly reversed

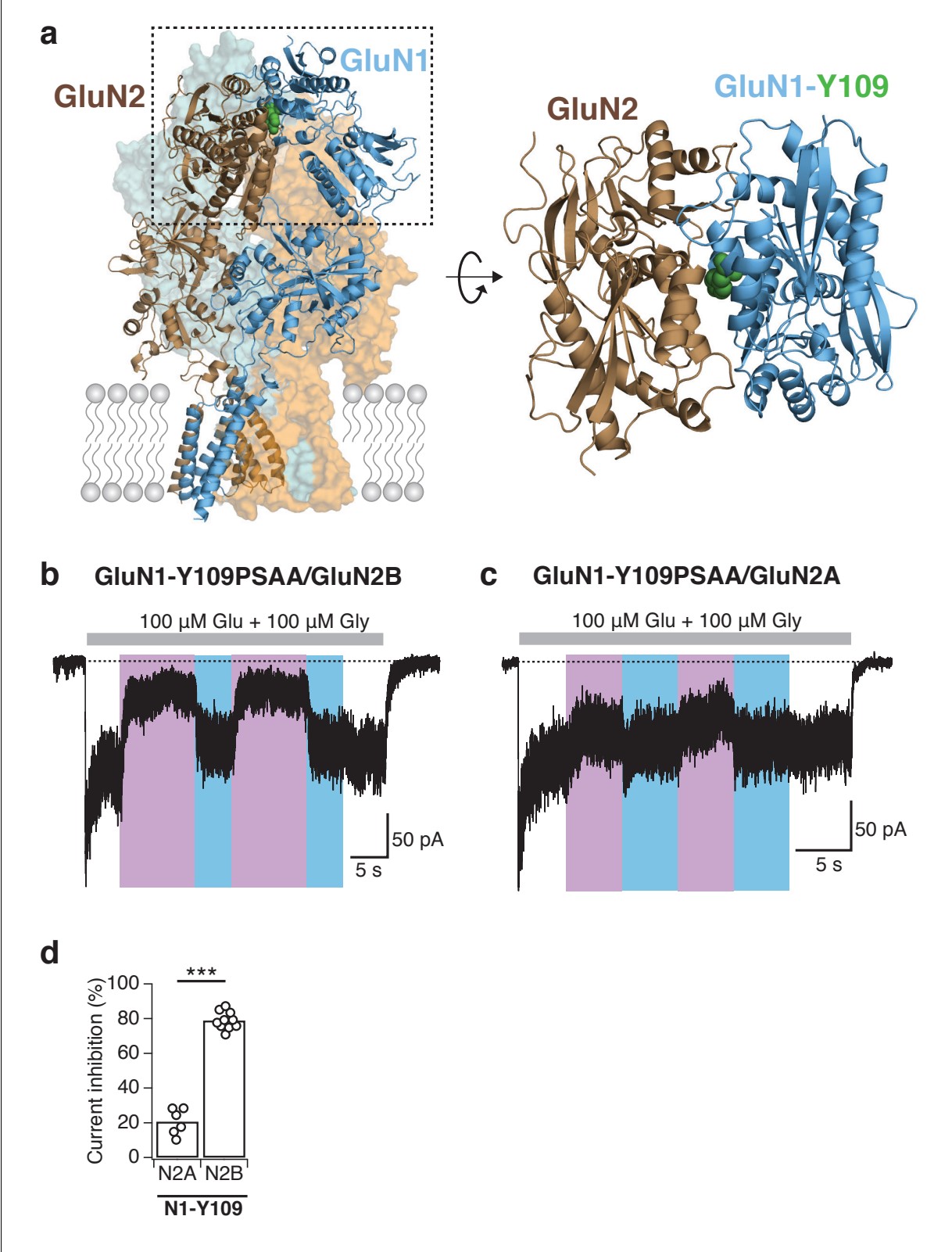

**Figure 5.** Reversible and subunit-specific photocontrol of GluN1-Y109PSAA/GluN2A receptors. (a) Structure of a heteromeric GluN1/GluN2B receptor (*left panel*) with PSAA inserted at the Amber mutation site GluN1-Y109 (*green spheres*) at the NTD heterodimer interface. The top view of one ABD dimer highlights the location of the PSAA insertion (*right panel*). (b) Representative current trace of GluN1-Y109PSAA/GluN2B receptors. In this example, the reduction of activity upon UV was 78%. (c) As in *b*, but for GluN1Y109PSAA/GluN2A receptors. Here, the UV-mediated photomodulation

*Figure 5 continued on next page*

*Figure 5 continued*

was only 28%. (d) Summary of peak current inhibition degrees upon exposure to UV for the GluN1-Y109PSAA mutant co-expressed with either GluN2A or GluN2B. On average, the photoinactivation (in %) is: $Inh_{GluN2B}$ = 79 ± 2 ($n$ = 10), $Inh_{GluN2A}$ = 21 ± 3 ($n$ = 6), ***p<0.001.

by shining blue light (*Figure 6b–e*). The photopotentiation observed for the TMD mutants contrasts the photoinhibition previously seen at NTD and ABD sites, thus demonstrating that PSAA-mediated receptor tuning can be bi-directional. The light-induced potentiation effects were particularly massive at GluN1-Y647 and GluN1-F654 sites (fold potentiation, 4.3 ± 0.4 [$n$ = 14] and 3.9 ± 0.2 [$n$ = 12], respectively), both fundamental parts of the M3 S<u>YTANLAAF</u> motif sequence (*Figure 6—figure supplement 1*). For these mutants, 'leakage' through unspecific Amber codon suppression was absent or minimal (*Figure 6—figure supplement 2*) and the extent of photopotentiation was similar for all states tested (with UV applied in the resting, active, or 'mixed' state; *Figure 6f*). This demonstrates that the *cis-trans* azobenzene interconversion occurs with similar efficacy irrespective of the receptor's functional state. Quantifying the on-relaxation kinetics of current potentiation, we observed marked site-dependence within the hydrophobic amino acid cluster (*Figure 6g*), indicating that the photopotentiation effect is strongly influenced by the local steric environment. Finally, in accordance with the previously described photoresponse properties at extracellular sites, the TMD photoresponses invariably displayed high (bi-) stability, full reversibility, and excellent repeatability (*Figure 6h and i*).

To dissect the mechanism underlying the UV-triggered photopotentiation effect, we then performed single-channel recordings on outside-out patches pulled from cells expressing either GluN1-Y647PSAA or GluN1-F654PSAA receptors, the two mutants showing the strongest UV-mediated effects. As a control, we first verified that WT GluN1/GluN2A single-channels properties (level of activity, conductance) were insensitive to light changes (dark, UV, blue; *Figure 7—figure supplement 1*). In accordance to the observations on the macroscopic level, the GluN1-F654PSAA mutant showed strong light responsiveness characterized by a massive increase of channel activity during UV exposure, with no or little change in single-channel conductance (*Figure 7a*). Quantification using all-points histograms showed that the unitary conductance was unaltered compared to WT channels (59.6 ± 3 pS [n = 4] for mutant *vs* 63.3 ± 2.4 pS [n = 5] for WT, p=0.38; *Figure 7—figure supplement 1*) and that the increase in single-channel activity ($N*P_o$) could account almost entirely for the current enhancement seen at the whole-cell level (*Figure 7c*). On the other hand, the behavior of the GluN1-Y647PSAA mutant was strikingly different. A predominant existence of the *trans*-isomer in the dark or blue state resulted in strongly disrupted single-channel properties, creating low-conductive and excessively brief channel openings. Although all-points amplitude histograms failed to identify well-defined single peaks, detailed inspection of the recordings consistently revealed a few rare openings suggestive of a diminished unitary conductance (although the channel may not have reached full amplitude because of filtering; *Figure 7b*). Flipping the azobenzene side chain into the bent *cis*-form upon UV caused a release from this constrained state as manifested by a sudden and dramatic increase in channel activity (*Figure 7c*). This large increase in activity was accompanied by the appearance of typical WT-like large conductance openings (*Figure 7b*). Whole-cell noise analysis further supported light-triggered modifications of the single-channel conductance of GluN1-Y647PSAA receptors, but not of GluN1-F654PSAA channels (*Figure 7—figure supplement 2*).

We gained further evidence that photoswitching GluN1-Y647PSAA directly impacts the channel permeation properties by investigating the photosensitivity to two well-characterized NMDAR pore blockers, $Mg^{2+}$ and MK-801. While GluN1-Y647PSAA receptors displayed similar $Mg^{2+}$-sensitivity as WT receptors under UV light ($IC_{50,UV}$ = 41.5 μM [$n$ = 4–10] *vs* $IC_{50,DARK}$ = 34.6 [$n$ = 4–7] for WT), their $Mg^{2+}$-sensitivity was significantly decreased under dark or blue light, as reflected by the 2.5-fold increase in $IC_{50}$ ($IC_{50,DARK}$ = 103 μM [$n$ = 4–6]; $IC_{50,BLUE}$ = 102 μM [$n$ = 3–7]; *Figure 7d* and *Figure 7—figure supplement 3a and b*). These effects were fully reversible, allowing rapid alternations between high and low $Mg^{2+}$-sensitivities in the same receptor population by simple PSAA toggling. On the other hand, $Mg^{2+}$ inhibited WT and GluN1-F654PSAA receptors with similar potency and in a light-independent manner (*Figure 7—figure supplement 3*). Because $Mg^{2+}$-inhibition is largely independent of channel activity (*Qian et al., 2002*), the light-induced changes in $Mg^{2+}$-sensitivity of GluN1-Y647PSAA mutants are likely attributable to changes in how $Mg^{2+}$-ions interact with the

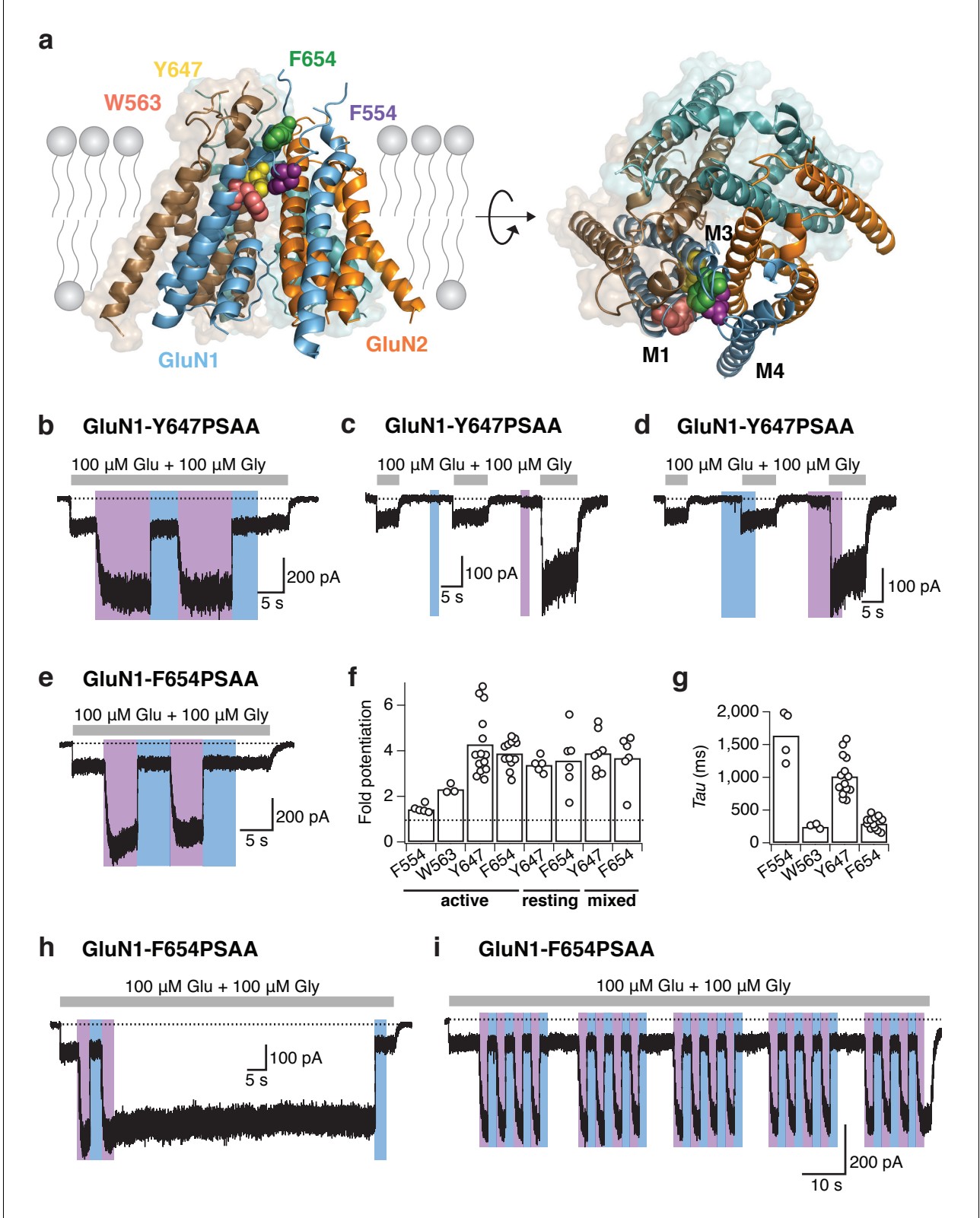

**Figure 6.** Reversible NMDAR photopotentiation with PSAA at various pore sites. (a) Structure of the GluN1/GluN2 TMD represented by a side (*left panel*) and top view (*right panel*). The four sites of the hydrophobic cluster, where PSAA was incorporated, are illustrated as spheres in different colors. (b) Example recording for GluN1-647PSAA/GluN2A mutants. UV illumination in the presence of the co-agonists resulted in a 3.4-fold potentiation of receptor activity. (c) Same cell as in *b*, with light applied in the resting state. Here, the UV-driven potentiation was 3.9-fold. (d) Same cell as in *b*, with

*Figure 6 continued on next page*

*Figure 6 continued*

light applied in the 'mixed' states. In this example, the current increased by 5-fold. (**e**) As is *b*, for the GluN1-654PSAA/GluN2A mutant. Here, UV light induced a 4.6-fold potentiation. (**f**) The fold-potentiation of activity differed between the mutants, but not between their functional states. On average, the photopotentiation in the active state is: $Pot_{F554} = 1.4 \pm 0.1$ ($n = 5$), $Pot_{W563} = 2.3 \pm 0.1$ ($n = 3$), $Pot_{Y647} = 4.3 \pm 0.4$ ($n = 14$), $Pot_{F654} = 3.9 \pm 0.2$ ($n = 12$). UV in the resting state resulted in $Pot_{Y647} = 3.4 \pm 0.2$ ($n = 5$) and $Pot_{F654} = 3.6 \pm 0.5$ ($n = 6$); and in the 'mixed' state in $Pot_{Y647} = 3.9 \pm 0.3$ ($n = 8$) and $Pot_{F654} = 3.7 \pm 0.5$ ($n = 6$). The dashed line indicates a potentiation of 1 (i.e. no potentiation). (**g**) Kinetics of photopotentiation. The mean photopotentiation exponential time constants are (in ms): $\tau_{F554} = 1640 \pm 170$ ($n = 4$), $\tau_{W563} = 240 \pm 20$ ($n = 3$), $\tau_{Y647} = 1020 \pm 80$ ($n = 14$), $\tau_{F654} = 290 \pm 30$ ($n = 8$). (**h**) Example trace showing that the UV-induced potentiation, once initiated, is highly stable even after UV light has been turned off (here, for 70 s). (**i**) Example trace demonstrating the stability of the photoresponse over numerous *trans-cis* illumination cycling events.

The following figure supplements are available for figure 6:

**Figure supplement 1.** Multiple sequence alignment in the TMD region across multiple iGluR family subunits (NMDA, AMPA, and kainate receptor subunits).

**Figure supplement 2.** No or minor unspecific Amber codon suppression at TMD sites.

pore. Similarly, inhibition of GluN1-Y647PSAA receptors by the large organic cation MK-801 revealed a pronounced photosensitivity. While under UV current the extent of inhibition by MK-801 (1 μM) was >85%, switching the azobenzene side chain to *trans* resulted in a marked reduction of MK-801 channel block (69% maximal inhibition; *Figure 7—figure supplement 4*). For GluN1-F654PSAA receptors, the degree of MK-801 block was also affected by light, although to a lesser extent, and reached values closer to WT levels (≥90%; *Figure 7—figure supplement 4c–f*). These results, combined with the single-channel and noise analysis data, indicate that the 'ease' with which ions flow through or block the GluN1-Y647PSAA channel pore can be manipulated by light. They also identify GluN1-Y647 as a novel residue critically involved in determining the energetics and size of the channel open state (see Discussion).

## Discussion

In this study, using a novel approach of chemical optogenetics, we have generated a family of NMDARs that can be accurately and reversibly controlled by light. We installed light sensitivity by genetically encoding azobenzene-containing UAAs that act as site-specific nanoscale photoswitches. We demonstrate that PSAA incorporation occurs with high fidelity and endows robust photoregulation combining full reversibility, high temporal precision, and molecular specificity. We identified several positions in different receptor regions that allowed to either photoinhibit or photopotentiate receptor activity through various mechanisms. Remarkably, these effects were driven by a minimal photoswitching event involving toggling between the elongated *trans-* and the bent *cis*-isoform of single azobenzene side-chains. The changes of receptor activity, detected on both the macroscopic and the single-channel level, were invariably reversible, sharply time-locked to the remote illumination events, and remarkably stable over long periods (minutes). By offering a novel approach to gain site-specific and reversible photocontrol on target proteins, our work significantly expands the repertoire of useful tools to engineer and manipulate biomolecules. Using this PSAA methodology, we were able to demonstrate its utility on studying NMDAR mechanisms, unveiling the contribution of specific side chains to distinct receptor properties such as agonist sensitivity, channel open probability, and permeation.

Reviewing our entire set of mutations, we found PSAA to be successfully incorporated in 50% of the cases (13/27 positions; *Table 1*), highlighting the feasibility and robustness of the methodology. PSAA azobenzene moieties are bulkier compared to natural substitutions as utilized in classical mutagenesis, nonetheless, our high rate of incorporation, as demonstrated by electrophysiologically detectable receptor functionality, argues for their overall good tolerance and comparable insertion to classical amino acid substitutions. Overall, the measured peak currents from the functional Amber mutant receptors tested were sufficiently large and stable, to enable long-lasting optical experiments (see *Figure 2—figure supplement 1*). Comparing the group of functional Amber mutants tested between each other, we have observed stop codon sites that were better tolerated, resulting

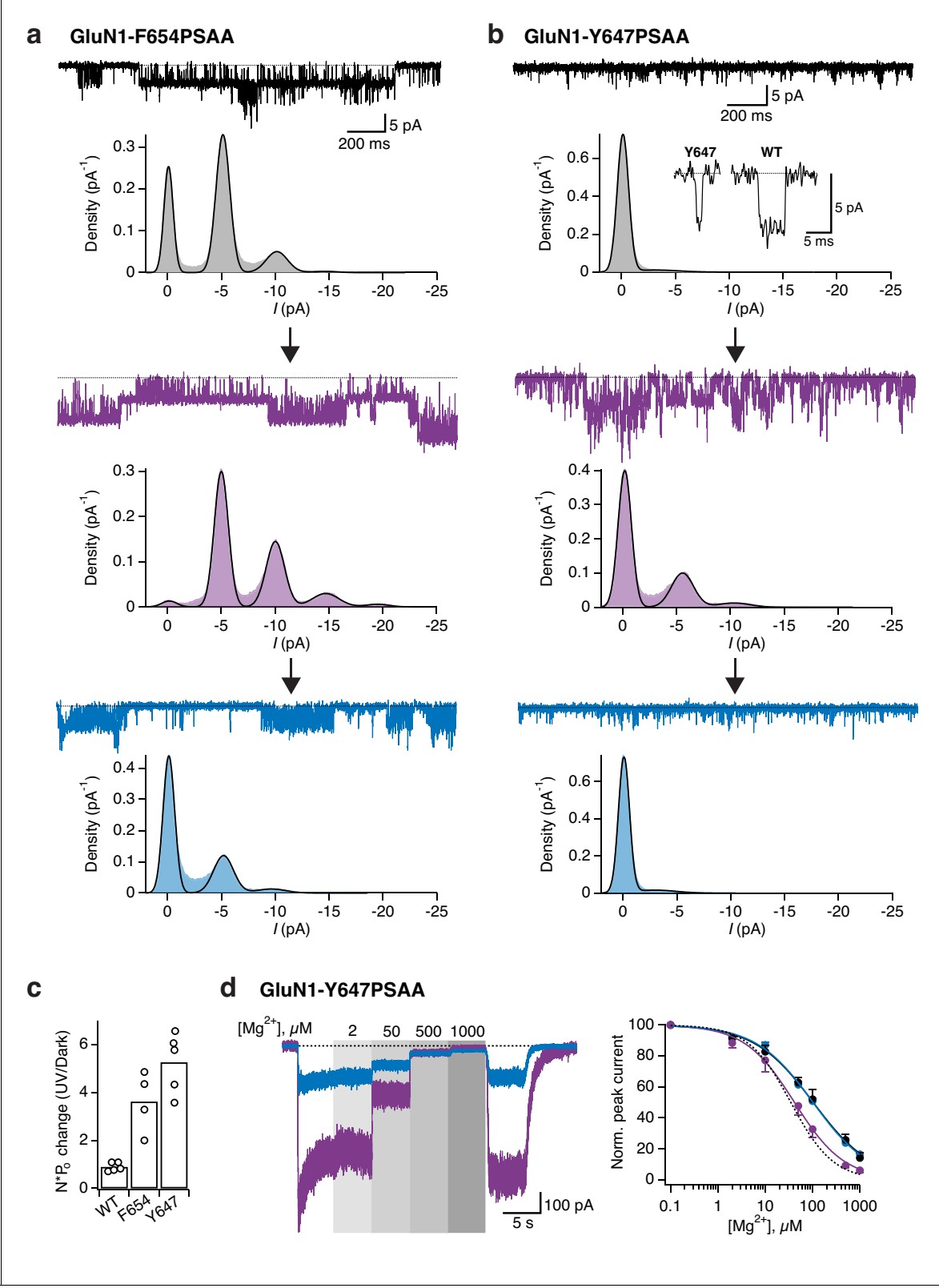

**Figure 7.** Optical modulation of receptor channel gating and permeation. (a) Example single-channel recording from an outside-out patch expressing GluN1-F654PSAA/GluN2A mutant receptors in the presence of 100 μM glutamate and glycine and exposed to different light conditions (dark, UV, blue). All-points amplitude histograms were produced from 11 s of total recording time for each light condition and fitted with multiple Gaussian components (*black lines*), allowing calculation of single-channel elementary conductance ($\gamma_{el}$) and of receptor activity (N*P$_o$). In this example, the values

*Figure 7 continued on next page*

*Figure 7 continued*

for $\gamma_{el}$ are (in pS): $\gamma_{el,DARK}$ = 64.3, $\gamma_{el,UV}$ = 66, $\gamma_{el,BLUE}$ = 65. Here, applying UV light resulted in a 2-fold increase in N*$P_o$ compared to the dark state. Data filtered at 2 kHz for illustration. (**b**) As in *a*, for the GluN1-Y647PSAA/GluN2A mutant. Glutamate concentration was 0.05 µM, glycine 100 µM. Values of $\gamma_{el}$ in UV is 69.4 pS. Here, applying UV light resulted in a 5.8-fold increase in N*$P_o$ compared to the dark state. *Top panel, inset*: example single-channel openings from WT and GluN1-Y647PSAA mutant in the dark. (**c**) Pooled change in N*$P_o$ between UV and dark state for WT and mutant receptors. Values are (from left to right): 0.9 ± 0.08 (*n* = 5), 3.6 ± 0.63 (*n* = 4) and 5.3 ± 0.56 (*n* = 5). (**d**) Example magnesium dose-response traces from GluN1-Y647PSAA/GluN2A receptors under UV or blue light (*left panel*). $Mg^{2+}$-DRCs under the three different light conditions (*right panel*). $IC_{50}$ values are (in µM): $IC_{50,DARK}$ = 103 (*n* = 4–6 for each point); $IC_{50,BLUE}$ = 102 (*n* = 3–7); $IC_{50,UV}$ = 41.5 (*n* = 4–10). The dashed line corresponds to the magnesium sensitivity of WT receptors in the dark.

The following figure supplements are available for figure 7:

**Figure supplement 1.** Wild-type GluN1/N2A receptors do not show any light-dependent modulation of single-channel properties.

**Figure supplement 2.** Estimation of the single-channel conductance by whole-cell noise analysis.

**Figure supplement 3.** External magnesium block is not affected by light for WT and GluN1-F654PSAA receptors.

**Figure supplement 4.** MK-801 inhibition kinetics of TMD mutant receptors.

in higher expression levels and comparable $P_o$ properties to wild-type channels (e.g. GluN1-P532). Other positions (e.g. those within the TMD hydrophobic cluster) showed a higher invasiveness by PSAA, producing ion channels with lower expression rates and disrupted channel open probabilities compared to wild-type receptors (all of these changes were photoisomerization state-dependent however). Generally, the genetically-encoded single-arm photoswitch, as provided by PSAA, facilitates the functional screening process, since it does not require residue proximity (as for cysteine disulfide bridges) nor any posttranslational protein labelling process (as for photoswitchable tethered ligands or PTLs).

Some of the Amber stop codon sites tested did not tolerate PSAA incorporation at all, as detected by the lack of agonist-induced currents (see *Table 1*). For instance, many GluN2 mutants did not lead to functional receptors, as detected by the lack of agonist-induced currents. In particular, sites within GluN2A and GluN2B homologous to the permissive GluN1-P532 or -Y535 positions at the ABD dimer interface were not tolerated. A hydrophobic pocket, located at the GluN2A ABD hinge, is required to stabilize this interface through intrusion of the GluN1-Y535 residue (*Furukawa et al., 2005*; *Yi et al., 2016*). This hydrophobic pocket is absent on the GluN1 subunit though. We thus speculate that introduction of a bulky hydrophobic PSAA in the GluN2 ABD hinge prevents proper receptor assembly by disrupting ABD dimerization through steric clashes. We note that PSAA introduction at two other extracellular positions within GluN2 (GluN2A-Y281 and GluN2B-Y282), both not situated at any inter-subunit contact sites or exposed interfaces, were tolerated and folded into functional receptors. Recapitulatory, the use of PSAAs, as shown in our study, extends the properties of classical mutagenesis with a remote, real-time and dynamic control of side-chain geometry, thus adding a critical novel dimension to protein mutagenesis.

PSAA introduction enabled rapid and reproducible light-dependent toggling of the receptor between two functional states. Importantly, addition of this new light-controllable allosteric switch occurred without compromising the overall receptor function, in particular its gating machinery. Therefore, PSAA provides an artificial control mechanism that is orthogonal to the natural one provided by evolution (ligand binding), an attractive feature for biological investigations. Introduction of the PSAA to the P532 Amber site in the GluN1 ABD hinge resulted in a pronounced, although incomplete (~50% inhibition at saturating agonist concentrations), receptor inhibition upon UV exposure. This effect was fully reversible with blue light, allowing temporary precise alternations between a fully and a partially activated state of the receptor. The ability to reversibly control the receptor channel activity by simple photoswitching of an azobenzene side chain located far from the ion channel pore provides a striking example of how highly localized atomic scale conformational changes can propagate through long distance to impact protein functionality. Our nanoscale photoswitch installed at the ABD dimer interface also echoes recent pharmacological studies showing that side

chain mobility in this region is instrumental in mediating action of positive and negative allosteric modulators and their respective stabilization of the active or inhibited receptor state (*Hackos et al., 2016*; *Yi et al., 2016*). A straightforward explanation of the partial nature of UV inhibition of GluN1-P532PSAA receptors is the presence of mixed subunits harboring either a PSAA or a natural amino acid, the later contributing to light-insensitiveness. This possibility could be excluded, however, based on our control experiments performed in the absence of the UAA showing negligible unspecific background (at the GluN1-P532 site and all other positions tested; *Figure 2—figure supplement 3* and *Figure 6—figure supplement 2*). Since the absorption spectra of the azobenzene *trans*- and *cis*-states overlap substantially (*Beharry and Woolley, 2011*; *Hoppmann et al., 2014*, *2011*), UV irradiation produces a photostationary state with <100% of the *cis*-isoform. This limited *cis*-isomer induction is also unlikely to account for the incomplete nature of UV inhibition of maximally-activated GluN1-P532PSAA receptors. Indeed, nearly 95% of photoinhibition was achieved when the glycine concentration was reduced to 1 μM (*Figure 4d*). This indicates that under our conditions where light duration and intensity are not limiting (*Figure 3—figure supplement 2*), the vast majority of the azobenzene side chains had switched to the *cis*-state following UV exposure, in good agreement with previous estimations with PSAA introduced into small model peptides (*Hoppmann et al., 2011*). Hence, the incomplete UV inhibition of GluN1-P532 receptors unlikely stems from an inefficient PSAA photochemistry. Rather, it likely finds its origin in the biological mechanism underlying the UV photoinhibition.

By studying two key gating parameters (channel activity, agonist sensitivity), we show that flipping the GluN1-P532PSAA azobenzene group to the *cis*-configuration triggers a drop in channel $P_o$ accompanied by a reduction in glycine, but not glutamate, sensitivity. Thus, at saturating glycine concentrations, only the decrease in channel $P_o$ impacts the receptor activity, resulting in 50% current inhibition. In contrast, at subsaturating glycine concentration, both effects are in play, resulting in greater current inhibition. The strong effect of the GluN1-P532 PSAA *trans-cis* isomerization on receptor gating further supports the critical importance of the ABD dimer interface in controlling NMDAR activity (*Furukawa et al., 2005*; *Gielen et al., 2008*; *Borschel et al., 2011*). It also reveals that PSAA at the GluN1-P532 site acts as a photomodulator of glycine binding. The strategic location of GluN1-P532 at the GluN1 ABD interlobe hinge is ideally suited to influence glycine binding by controlling GluN1 ABD clamshell conformational dynamics. We propose that the PSAA in its *trans*-state (dark and blue condition) stabilizes the GluN1 ABD in a closed-cleft (i.e. active) conformation. Induction of the *cis*-isoform destabilizes this active conformation, favoring clamshell opening, which in turn accelerates glycine dissociation (*Figure 8a and b*). This proposed allosteric mechanism bears striking resemblance with the mode of action of TCN compounds, a family of negative allosteric modulators of glycine binding that bind the GluN1/GluN2A ABD dimer interface and make direct atomic interactions with several GluN1 ABD hinge residues including GluN1-P532 (*Hansen et al., 2012*; *Hackos et al., 2016*; *Yi et al., 2016*). At glutamatergic synapses, NMDAR activity is strongly dependent on the level of co-agonists (glycine, D-serine) in the receptor's vicinity. Until now, the exact physiological co-agonist concentrations at synaptic sites as well as their dynamic changes during neuronal activity are unknown (*Mothet et al., 2015*). We envision using the glycine dependence of GluN1-P532PSAA receptor photomodulation as an original means to assess glycine levels in the synaptic cleft. Designing a PSAA-containing GluN2 subunit in which glutamate sensitivity could be manipulated would provide another promising optochemical tool to control synaptic strength in a subunit-dependent manner by simple toggling between *cis*- and *trans*-configurations.

Exploiting the key advantage of UAA site tolerance, we identified a cluster of conserved aromatic residues in the TMD region that allowed to photocontrol the receptor's ion channel behavior. These residues, situated in the upper part of the TMD, form a 'hydrophobic cluster' facing opposite to the channel lumen and strategically connecting the GluN1 channel gate region (M3) to the peripheral GluN1 M1 helices. Placing the PSAA at sites within this patch resulted in receptors whose activity could be *enhanced* by light. The photopotentiation was particularly pronounced at GluN1-Y647 and -F654, two M3 residues sitting directly at the back of the channel gate. Interestingly, despite their similar extent of photopotentiation, the two mutants showed a remarkable difference in their macroscopic current noise level in the photopotentiated state, which was noticeably larger for GluN1-Y647PSAA receptors (compare current traces in *Figure 6b and e*). Our single-channel recordings reveal that this effect likely originates from the dual increase of channel $P_o$ and conductance for the GluN1-Y647PSAA variant, while only channel $P_o$ is affected for the GluN1-F654 mutant. Thus, the

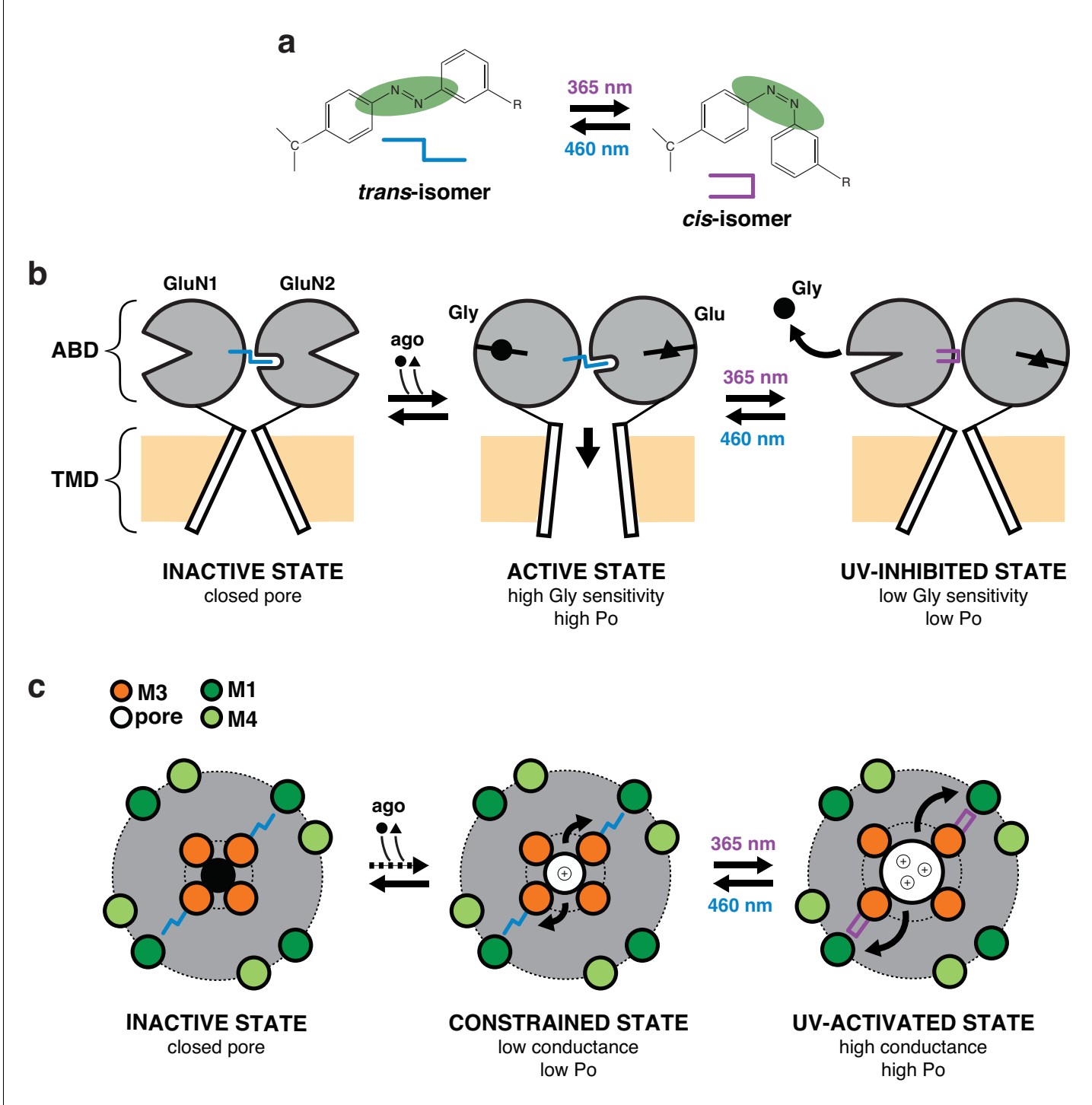

**Figure 8.** Proposed mechanism for photomodulation of the gating and permeation properties of NMDARs carrying PSAAs. (**a**) Toggling of the PSAA azobenzene moiety between the *trans*- and *cis*-isomers in dependence of blue or UV light. For simplicity, the two isomers are replaced by symbols as indicated. (**b**) Schematic representation of the mechanism underlying the photomodulation at the ABD level. For clarity, only one dimer is represented and the NTD region is omitted. PSAA in its *trans*-configuration (dark and blue condition), when placed at the GluN1 ABD clamshell hinge (P532 position), intrudes into a hydrophobic pocket located at the GluN2A ABD hinge region (*Furukawa et al., 2005*). This interaction stabilizes the GluN1 ABD in a closed-cleft conformation (i.e. active state) as well as ABD GluN1/GluN2 heterodimer interface, resulting in active (i.e. open) channels (*Furukawa et al., 2005*; *Gielen et al., 2008*). Flipping the GluN1-azobenzene moiety to the *cis*-configuration by UV illumination destabilizes this active conformation, favoring GluN1 clamshell opening, which in turn accelerates glycine dissociation. This effect is accompanied by a weakening of the ABD dimer interface resulting in a 50% drop of channel Po (UV-inhibited state). (**c**) Schematic representation of the mechanism underlying photomodulation

*Figure 8 continued on next page*

*Figure 8 continued*

in the pore domain. The TMD pore region is viewed from the top. PSAA is placed at the GluN1-Y647 site, which is part of a highly conserved hydrophobic cluster that sits at the back of the channel lumen and connects the M3 helix bundle (channel gate, inner ring) to the peripheral M1/M4 helix outer ring. PSAA in its extended *trans*-configuration prohibits adequate channel opening by preventing full expansion of inner M3 ring towards the periphery. The channel is thus locked in a constrained 'open' state, highly unstable (low $P_o$) and with low conductance. Upon UV, however, activity can be drastically enhanced, an effect that originates from the relief of the steric hindrance allowing the inner M3 ring to accomplish its full movement towards the outer ring (UV-activated state). This motion generates an increase in ion channel $P_o$ and conductance. Thus, channel opening is energetically more favorable in the more compact *cis*-configuration than the rigid and elongated *trans*-configuration of the PSAA.

GluN1 hydrophobic cluster emerges as a novel structural determinant that controls the energetics of channel gating. Although the structural basis of the *cis-trans* effects on channel $P_o$ remains to be elucidated, we speculate that steric hindrance plays a major role. During channel gate opening, the hydrophobic cluster is likely compressed to accommodate the helical packing due to the expansion (*Kazi et al., 2013*; *Tajima et al., 2016*) of the inner M3 bundle towards the M1/M4 outer ring. Accordingly, the rigid and elongated *trans*-configuration of the PSAA renders channel opening energetically less favorable than the smaller *cis*-configuration, thus accounting for the current increase when shining UV light. We speculate that in the case of GluN1-Y647PSAA, which locates deeper in the ion channel pore than GluN1-F654PSAA, steric hindrance imposed by the extended *trans*-isomer might even have more dramatic effect by preventing full expansion of the M3 bundle. On one hand, this strong structural constraint would greatly decrease the stability of the channel open state, accounting for the very brief openings. On the other hand, ion permeation would become less favorable, thus resulting in the decreased single-channel conductance and pore block effects. Probably, photoswitching to the *cis*-state relieves much of this constraint allowing the channel to fully dilate and regain its 'large' WT-like 60 pS conductance (*Figure 8a and c*). These results demonstrate the first use of single photo-active amino acids to directly control transmembrane domain motions and ion channel properties. The M2 loop, which sits deep in the pore and forms the region of highest constriction, is well known to have critical role in determining single-channel conductance and $Mg^{2+}$-block of NMDAR channels (*Wollmuth et al., 1998*; *Siegler Retchless et al., 2012*). Our data now reveal that the upper M3 channel gate region can also participate in controlling ion permeation. Finally, our data point to the transmembrane cavity between the inner M3 ring and the outer M1/M4 ring as a site with strong modulatory potential. A number of allosteric modulators targeting glutamate receptors have been proposed to bind the TMD region of glutamate receptors outside of the channel lumen (where channel blockers act) (*Traynelis et al., 2010*; *Zhu and Paoletti, 2015*). The transmembrane cavity described here represents an ideal locus to accommodate such ligands and fine-tune receptor activity.

With its site flexibility, reversibility, and genetic encodability, we anticipate that the PSAA nanoswitch approach described here will prove useful in native settings to optically manipulate neuronal proteins and advance our understanding of the molecular basis of brain function. With the high spatiotemporal precision conferred by light, we are able to control receptors on a time scale (ms to s) compatible with their biological activity, and in regard of NMDARs, with their rapid activity-dependent changes in subunit composition, which can occur on a time scale of minutes (*Sanz-Clemente et al., 2013*). Also, the genetic encodability affords high molecular (i.e. subunit) specificity and cell-type targeting. Finally, by involving allosteric mechanisms, rather than direct orthosteric interactions as for photoswitchable tethered ligands (PTLs; *Reiner et al., 2015*), the PSAA approach provides the ability to fine-tune receptor activity without interfering with its natural activation. Thus, highly accurate optical manipulation of NMDAR-mediated excitatory signals in defined neuronal circuits should not only enable to resolve the contributions of the two prominent subunits GluN2A and GluN2B to synaptic plasticity, but also of the less common GluN2C and GluN2D subunits. For instance, the photocontrol of GluN2A or GluN2B glutamate deactivation kinetics by PSAAs could assess how this gating parameter sculpts the time window for synaptic plasticity and integration. Similarly, we envision evaluating the contribution of GluN2D ultra slow deactivation kinetics (*Vicini et al., 1998*; *Vance et al., 2012*) to neuronal excitability, still an unresolved issue. 'On-demand' facilitation or inhibition of specific NMDAR subpopulations should also provide critical insights into how NMDAR subunits influence opposite forms of synaptic plasticity (LTP and LTD; see

*Yashiro and Philpot [2008]*). By allowing acute and reversible silencing of specific receptor entities, we expect the PSAA approach to surpass the classical gene knock-out experiments both by its temporal accuracy and lack of compensatory mechanisms. PSAAs should also be well applicable in interrogating the reversing of NMDAR hypofunction in animal disease models. Reduced NMDAR signaling is associated with several neurological disorders, including schizophrenia and cognitive impairments (*Paoletti et al., 2013*; *Yuan et al., 2015*; *Burnashev and Szepetowski, 2015*). Accordingly, there is currently great interest in developing methodologies to boost GluN2A or GluN2B function specifically to determine whether enhancing one pool of receptors, or both, leads to beneficial outcomes. Finally, we foresee great potential for PSAAs, and UAAs in general, for studying intracellular receptor mechanisms and synapse biology. With their major advantage of site tolerance, PSAAs can be incorporated at specific sites in the subunit cytoplasmic tails, which may allow to gain optical control on the receptor association with scaffolding proteins or signaling partners. Real-time photo-manipulation of the synaptic anchoring of NMDARs or their coupling to key cellular messengers (e.g. GluN2B-CamKII; *Sanz-Clemente et al., 2013*) would certainly represent a major technical advance for clarifying the contentious roles (*Parsons and Raymond, 2014*; *Zhou et al., 2015*) of NMDAR subunit trafficking and signaling in synaptic plasticity and neuronal survival. The instantaneous and reversible on-off photoswitch provided by PSAAs is bound to afford superior spatiotemporal control of the receptor intracellular interactome compared to conventional approaches based on slowly-acting interfering ligands or genetic modifications.

Implementing optochemical approaches to study brain proteins in native situations is challenging (*Kramer et al., 2013*), but feasibility is within reach, be it for photosensitive ligands (*Szobota et al., 2007*; *Caporale et al., 2011*; *Pittolo et al., 2014*; *Lin et al., 2015*; *Laprell et al., 2015*; *Berlin et al., 2016*; *Levitz et al., 2016*) or UAAs (*Ernst et al., 2016*; *Han et al., 2017*; *Zheng et al., 2017*; *Chen et al., 2017*). There are two main requirements for in vivo incorporation of UAAs in mammals: (*i*) efficient delivery and expression of the genes encoding the orthogonal tRNA/synthetase pair and the mutated target protein of interest; and (*ii*) sufficient bioavailability of the UAA at the desired tissue or cell type. Successful attempts to address these challenges have recently emerged. *In utero* electroporation of plasmid DNAs coupled to direct injection of the UAA into the embryonic brain provides a first possible route (*Kang et al., 2013*). Gene transfer using adeno-associated viral (AAV) vectors offers an alternative way for efficient delivery of the UAA genetic machinery in mammalian cells, tissues, and brains of living mice (*Ernst et al., 2016*; *Zheng et al., 2017*). The generation of transgenic mice with a heritable expanded genetic code - i.e. mice directly incorporating the necessary tRNA/synthetase genes for Amber stop codon suppression in their genome - is the most recent development in the UAA field, which will certainly greatly facilitate in vivo UAA applications in the future. Mouse strains allowing incorporation of the photocrosslinker AzF (*Chen et al., 2017*) or the post-translationally modified lysine $N^\varepsilon$-acetyl-lysine (AcK; *Han et al., 2017*) have just become available. We expect this strategy to be extended to other tRNA/synthetase pairs in a custom-made manner, including the pair utilized in the current study enabling site-specific incorporation of PSAAs. UAA delivery in the whole animal can be achieved by intraperitoneal injection, or, directly into target tissues for spatio-specific induction of UAA-engineered protein expression (*Han et al., 2017*). UAA supplementation in the mouse drinking water provides another easy-to-use and viable option, even when targeting brain proteins as recently shown with a lysine derivative (*Ernst et al., 2016*). Depending on the UAA chemistry, intracranial UAA perfusion (also performed by *Ernst et al., 2016*) may be required to reach sufficient UAA levels in neural cells. The bioavailability of PSAAs is currently unknown, but in case of poor brain-blood-barrier crossing, direct brain injection using a multimodal optogenetic cannula for dual drug and light delivery (*Canales et al., 2015*) would advantageously combine PSAA supplementation with its light-controlled photoisomerization. The recent development of red-shifted azobenzene derivatives (*Samanta et al., 2013*; *Kienzler et al., 2013*), including a PSAA version controllable by visible light (*Hoppmann et al., 2015*), should further enhance biocompatibility and offers promising perspectives for in vivo applications of azobenzene-based photoswitches.

## Materials and methods

### Molecular biology

The rat GluN1-1a (named GluN1 herein), rat GluN2A, and mouse GluN2B subunits were expressed from pcDNA3-based vectors. Site-specific Amber codon (TAG) mutations for the insertion of the photoswitchable amino acid (PSAA) were introduced by means of site-directed Quikchange mutagenesis and confirmed by sequencing, as previously described (*Zhu et al., 2014*). A list of all Amber mutations tested in this study is given in *Table 1*. The incorporation of PSAA was driven by an orthogonal tRNA/synthetase pair, derived from a *Methanosarcina mazei* (*Mm*) pyrolysine tRNA/synthetase pair (tRNA$^{Pyl}$–*Mm*PSCAA-RS) and specifically engineered for suppressing the introduced stop mutation (*Hoppmann et al., 2014*). The designed synthetase recognized exclusively the *trans*-version of the PSAA, which is also the thermodynamically more stable isoform in the dark state. Wild-type eGFP or eGFP carrying an Amber mutation at the Y37 site both served as transfection controls.

### Cell culture, transfection, and UAA incubation

Wild-type and mutant NMDARs were expressed in HEK-293 cells (obtained from ATCC Inc.). The cells were incubated in DMEM containing 10% FBS and 1% Penicillin/Streptomycin (complete medium). The transfection of NMDAR subunits was performed using polyethylenimine (PEI) in a DNA/PEI ratio of 1:3 (*v/v*), as previously described (*Klippenstein et al., 2014*). For recordings of heteromeric NMDARs, the following vectors were co-transfected: (*i*) a WT subunit (either GluN1, GluN2A, or GluN2B); (*ii*) an Amber mutant subunit (with a TAG codon within GluN1, GluN2A, or GluN2B); (*iii*) the tRNA/synthetase pair; and (*iv*) eGFP or eGFP-Y37TAG as a control of transfection. Typically, the total amount of DNA was 1.5 µg per 0.8 cm cover slip and the mass ratio was 1:2:1:1. PSAA was obtained from two different sources: (*i*) in-house chemical synthesis at UCSF (laboratory of Lei Wang), and (*ii*) customized synthesis by Enamine Ltd. (Kiev, Ukraine). No difference in photoregulation was observed between either compound source. Stock solutions of 30–50 mM PSAA were prepared by dissolving the PSAA through sonication in 0.1 N NaOH. Complete medium, supplemented with the final PSAA concentration of 0.3–0.5 mM (pH 7.3) and 150 µM of the competitive antagonist D-APV (to avoid cell toxicity caused by receptor overexpression), was added immediately after transfection.

### Electrophysiology and photomodulation

#### Whole-cell recordings and pharmacology

Receptor functionality and light-sensitivity were assessed in patch-clamp recordings of lifted whole cells 24–72 hr post transfection. Positively transfected cells were visualized by GFP-fluorescence. The patch pipettes had a resistance of 4–6 MΩ and were filled with a solution containing (in mM): 115 CsF, 10 CsCl, 10 HEPES and 5 EGTA (pH 7.2 with CsOH). The osmolarity was 270–290 mOsm. The cells were continuously perfused with a Ringer solution containing (in mM): 140 NaCl, 2.8 KCl, 1 CaCl$_2$, 0.01 DTPA, 10 HEPES and five sucrose (pH 7.3 with NaOH). The osmolarity was 290 mOsm. DTPA was added to prevent tonic inhibition of NMDARs by ambient zinc (*Paoletti et al., 1997*). Currents were recorded using an Axopatch 200B amplifier and a Digidata 1550A interface (Molecular Devices), sampled at 10–20 kHz, and low-pass filtered at 2–5 kHz. The cells were voltage-clamped at −60 mV. pClamp 10.5 (Molecular Devices) was used to acquire the data. All recordings were performed at room temperature.

Agonists or other pharmacological substances were applied using a multi-barrel rapid solution exchanger system (RSC 200, Bio-Logic). Typically, NMDARs were activated by application of 100 µM glutamate on a background of 100 µM glycine. Glutamate and glycine dose-response curves (DRCs) were performed in the presence of 100 µM of the respective co-agonist. The agonist deactivation kinetics were assessed by including the converse co-agonist to the control solution. For Mg$^{2+}$-DRCs, the concentration of DTPA, which has orders of magnitude higher affinity for Zn$^{2+}$ than Mg$^{2+}$ (*Paoletti et al., 1997*), was reduced to 1 µM to avoid excessive Mg$^{2+}$ chelation, while maintaining depletion of ambient zinc (estimated in the few hundreds of nM). Accordingly, Mg$^{2+}$ concentrations reported in the Mg$^{2+}$-DRC were not corrected and correspond to the nominal concentrations of added Mg$^{2+}$. MK-801 was applied at 1 µM to assure full inhibition of WT NMDAR-mediated

currents. The positive allosteric modulator (PAM) GNE-6901 (obtained from Genentech, South San Francisco, USA) was used at a final concentration of 30 μM (*Hackos et al., 2016*).

## Single-channel recordings

Single-channel currents of outside-out patches expressing TMD-Amber mutant or wild-type receptors were recorded 12–24 hr post transfection. Patch pipettes had a resistance of 10–12 MΩ and were filled with the same solution as for whole-cell recordings. The external perfusion solution was identical to that used in whole-cell recordings except that $CaCl_2$ concentration was lowered to 0.5 mM in order to minimize the appearance of NMDAR channel sub-conductance states (*Premkumar et al., 1997*). NMDARs were activated by applying saturating glycine (100 μM) and variable glutamate concentrations (100, 0.05, or 0.02 μM), depending on the level of channel activity. Recordings were performed at −80 mV. The currents were acquired at a sampling rate of 20–50 kHz and low-pass filtered at 2 kHz.

## Photomodulation

Computer-controlled light pulses during electrophysiological recordings were provided from sensitive high power LEDs (Prizmatix). The three following LEDs were used: Mic-LED-365 (UV, 200 mW), UHP-Mic-LED-460 (blue, 2W) and UHP-Mic-LED-520 (green, 900 mW). The LED port was directly coupled via a microscope adaptor to the fluorescence port of an inverted DIAPHOT 300 Nikon microscope. The output beams of the three Mic-LEDs were joined by a dichroic beam combiner module and the resulting light was applied to the center of the recording dish through a 10X objective (Leitz Wetzlar). The light intensity of all three LEDs was set to 100%, unless otherwise noted. Pulses of UV, blue, or green light had a minimal duration of 100 ms.

The UV/Vis spectra of the free PSAA (100 μM) were measured using a UVIKON 922 spectrophotometer (Kontron Instruments) in 1 x 1 × 4 cm quartz cuvettes. The spectra were measured at two different conditions: (*i*) with PSAA in the extracellular (Ringer) solution to be as close as possible to the recording conditions, and (*ii*) in isopropanol to mimic a hydrophobic environment (since PSAA is buried in the protein) (*Ye et al., 2009*). To induce the different photoisomers, the dissolved PSAA was directly irradiated in the cuvette at 365 nm (30 mW), 460 nm (241 mW), and 525 nm (35 mW) using a Cool LED pE-4000 illumination system. The UV spectra of the pure *trans*-isomer were recorded upon storage in the dark.

## Analysis

Current traces obtained in electrophysiological recordings were transferred to Igor Pro (version 6.3) or Clampfit (version 10.5) for display and analysis. The kinetics of photoinactivation and photopotentiation were obtained by fitting currents with a single exponential function as follows: $Y = A*exp(-t/\tau) + C$, with $A$ as the initial current amplitude, $Tau$ ($\tau$) the decay time constant, and $C$ the steady-state level. The inhibition by MK-801 was also fit with a monoexponential function. Although for WT receptors and a subset of mutant receptors (e.g. the GluN1-P532PSAA mutant), double exponential functions yield better fits of the MK-801 inhibition (as previously described *Buck et al. [2000]*), for the tested TMD mutant receptors with very slow rates of MK-801 channel block (especially in the dark state), double component fits were unable to provide interpretable values. Accordingly, and for comparative purposes, only single exponential fits were used when studying MK-801 inhibition. Agonist DRCs were fit with the following Hill equation: $I_{rel} = 1/(1 + (EC_{50} / [ago]^{nH})$, with $EC_{50}$ and the Hill coefficient ($n_H$) as free parameters. For $Mg^{2+}$-DRCs the following equation was used: $I_{rel} = 100*(1-(a/(1+[IC_{50}]/[Mg^{2+}]^{nH},)))$, where $a$ is the maximal current inhibition by $Mg^{2+}$. $IC_{50}$, $a$, and the Hill coefficient ($n_H$) were set as free parameters.

Glutamate and glycine deactivation responses were fitted with a double exponential function with a fast and a slow component as previously described *Furukawa et al. (2005)*: $Y = A_f*exp(-t/\tau_f) + A_s*exp(-t/\tau_s)$, with $A_f$ and $A_s$ as the amplitudes and $\tau_f$ and $\tau_s$ as the decay time constants of the fast and slow decay components, respectively. The weighted deactivation time constants were calculated with: $\tau_w =( A_f/(A_f+A_s))*\tau_f + (A_s /(A_f + A_s))*\tau_s$.

For the analysis of whole-cell noise, currents were sampled at 20 kHz and low-pass filtered at 10 kHz. Noise analysis was performed from segments of at least 6 s of steady currents in the dark and under UV illumination ($\geq$6 s each, same cell). The apparent single-channel conductance was

estimated from the whole-cell noise using the following equation (*Cull-Candy et al., 1988*; *Smith et al., 1999*): $\gamma_{noise} = var(I)/[(m_I)*(1-P_o)*(\Delta V)]$, where $m_I$ is the mean current amplitude evoked by the agonists, $P_o$ the receptor channel open probability, $\Delta V$ the transmembrane voltage driving force (60 mV, with a holding potential of $-60$ mV and assuming a reversal potential of 0 mV), and var(I) the variance of the current around the mean. $P_o$ values in the dark, for mutant (mut) GluN1-F654PSAA and GluN1-Y647PSAA receptors, were estimated from MK-801 inhibition kinetics obtained in the current work: $P_o (mut,dark) = [P_o (wt)]/[ \tau_{on,MK801} (mut)/\tau_{on,MK801} (wt)]$, where $P_o (wt)$ is the maximal channel open probability of wild-type (wt) GluN1/GluN2A receptors (0.5; *Paoletti, 2011*). $P_o$ values under UV illumination were calculated as follows: $P_o (mut,UV) = m_I (UV)/m_I (dark)$. This ratio was determined for each individual cell. Relative changes in $\gamma_{noise}$ (R$\gamma$) between UV and dark were then calculated, for each individual cell, using the formula: $R\gamma = \gamma_{noise} (UV)/\gamma_{noise} (dark)$.

Statistical significance was assessed with Student's *t*-test, using either pairwise comparisons for different values from a single cell, or unpaired tests for comparisons between different mutants or cells. Error bars represent the standard deviation of the mean value (SD).

For structural representations of the receptor, a homology model was built using the two full-length structures pdb 4PE5 (*Karakas and Furukawa, 2014*) and pdb 4TLM (*Lee et al., 2014*) as templates.

## Acknowledgements

This work was supported by the Fondation pour la Recherche Médicale ('Equipe FRM' grant #DEQ2000326520 to PP), the French government ('Investissements d'Avenir' ANR-10-LABX-54 MEMO LIFE, ANR-11-IDEX-0001-02 PSL* Research University, and ANR JCJC ChemProbe to SY), the European Research Council (ERC Advanced Grant #693021 to PP), the Deutsche Forschungsgemeinschaft (DFG; postdoctoral fellowship #KL 2935/1-1 to VK), and by the U.S. National Institutes of Health (1R01GM118384-01 to LW). We thank Laetitia Mony and Laura Piot for measuring the UV/Vis spectra of PSAA and David Stroebel for generating the model of the GluN1/GluN2B receptor. We also thank Laetitia Mony for critical reading of the manuscript. The authors declare no competing financial interests.

## Additional information

### Funding

| Funder | Author |
| --- | --- |
| Agence Nationale de la Recherche | Viktoria Klippenstein<br>Shixin Ye<br>Pierre Paoletti |
| Centre National de la Recherche Scientifique | Pierre Paoletti |
| Institut National de la Santé et de la Recherche Médicale | Shixin Ye<br>Pierre Paoletti |
| National Institutes of Health | Christian Hoppmann<br>Lei Wang |
| European Research Council | Pierre Paoletti |
| Fondation pour la Recherche Médicale | Pierre Paoletti |
| Deutsche Forschungsgemeinschaft | Viktoria Klippenstein |

The funders had no role in study design, data collection and interpretation, or the decision to submit the work for publication.

## Author contributions

VK, Conceptualization, Formal analysis, Funding acquisition, Validation, Investigation, Visualization, Methodology, Writing—original draft, Writing—review and editing; CH, Resources, Methodology, Writing—review and editing; SY, Conceptualization, Resources, Funding acquisition, Methodology, Writing—review and editing; LW, Resources, Funding acquisition, Methodology, Writing—review and editing; PP, Conceptualization, Resources, Formal analysis, Supervision, Funding acquisition, Validation, Investigation, Visualization, Methodology, Writing—original draft, Project administration, Writing—review and editing

## Author ORCIDs

Lei Wang, http://orcid.org/0000-0002-5859-2526

Pierre Paoletti, http://orcid.org/0000-0002-3681-4845

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
