## [Decision Letter]

Thank you for submitting your article "Optocontrol of glutamate receptor gating and permeation by single side-chain photoisomerization" for consideration by *eLife*. Your article has been reviewed by three peer reviewers, and the evaluation has been overseen by a Gary Westbrook as the Senior Editor and Reviewing Editor. The following individuals involved in review of your submission have agreed to reveal their identity: Stephen Traynelis (Reviewer #2); Shai Berlin (Reviewer #3). The reviewers have discussed the reviews with one another and the Senior Editor has drafted this decision to help you prepare a revised submission.

Summary of Editor/Reviewer discussion:

This is a beautifully written paper but also potentially a niche paper, which in time could be influential after further technical development. At this stage we think the authors should revise the manuscript within the guidelines of the Tools and Resources section of *eLife*. As it stands it does not advance glutamate receptor biology now, but it is a demonstration of a new tool that could change how one designs experiments. One hurdle is the ability to encode the unnatural amino acid photoswitch in vivo, thus a thoughtful discussion by the authors of what exactly is needed to apply this technology in vivo would increase the value of the manuscript. It was smart to initially target positions that were found to influence receptor function (e.g. P532, Furukawa et al. Nature 2005), serving as a reference. Notably, the mutagenesis of these residues in previous reports is not analogous to what is shown here: mutations are static, whereas the UAAs incorporated here are not. If the authors discussed these differences more in depth it would add a stronger biological perspective to the paper. It would also be useful if the authors could provide examples for what sorts of question could be answered using photo switchable NMDA receptors, that have not already been addressed using conventional techniques. These general comments as well as the specific comments of the reviewers listed below should be used to guide the revision.

*Reviewer #1:*

This paper reports an exciting advance in photobiology achieved by combining two existing technologies which have been previously applied to study glutamate receptors: Genetic incorporation of unnatural amino acids (UAA), and azobenzene photoswitching of side chain conformation. In combination, this allows light to trigger reversible switching between two functional states of the receptor. The use of azobenzine UAA derivatives is an advance over prior work in which photo switches were introduced by chemical modification of mutant receptors containing a thiol (Cys) at the targeted position, in part because only solvent accessible residues can be targeted with thiol reactive photo switches, while with UAA incorporation buried residues can be targeted, provided that the UAA side chain does not interfere with protein expression and folding or receptor function. The data is very high quality and the paper well written, albeit with over interpretation of what is actually learned with this approach.

However, beyond demonstration of proof of principle, the paper does not greatly advance our understanding of receptor function. Two of the sites, in the ligand binding domain dimer interface, and in the amino terminal domains were previously demonstrated to be important regulators of receptor activity, and UAA photo switching yields no new information about receptor function. The third site, in a hydrophobic cluster near the gate of the ion channel domain, yields receptors with light gated changes in single channel open probability, and very subtle changes in channel block which the authors over interpret as a change in permeation properties, consistent with light triggered structural changes in the ion channel domain, but again the results give no new insight into receptor function or the underlying structural changes; in a sense the approach is not much of an advance over traditional mutagenesis in that it does not yield structurally informative insights into channel function. An additional caveat is that the size and chemistry of the labeled residue introduced by UAA incorporation is hugely different from that of the wild type side chain, and thus in either of the light triggered conformations, receptor function is non-native.

A final caveat is that the UAA azobenzene approach will likely be difficult, perhaps impossible, to use in vivo and if this is true, the elegant recordings from transiently transfected HEK cells will have little impact beyond biophysical studies on recombinant receptors. Much of the excitement in optogenetics comes from the ability to alter receptor function in intact tissues, and in behaving animals. This is a beautiful paper, but the approach is a niche study which I think will not have great impact on the field.

*Reviewer #2:*

Work presented in this manuscript from Klippenstein et al. uses a novel approach to interrogate NMDA receptor function. By engineering optocontrollable NMDA receptors through the introduction of photoswitchable amino acids, the authors were able to identify positions within GluN1, GluN2A, and GluN2B at which conformational changes in a cis/trans isomerization of azo-benzene will alter receptor function, agonist potency, and channel gating. Specifically, GluN1-Y109 in the NTD, GluN1-P532 in the ABD, and GluN1-F554, W563, Y647, and F654 in the TMD showed notable photosensitivity. This is certainly an interesting application of a relatively new technology to the exploration of the structure-function relationship for ion channels. Specifically, the introduction of an azo-benzene side chain that can isomerize between cis and trans in response to different wavelengths of light allows real-time perturbation of protein structure, and thus allows new kinds of experiments to be conceptualized and carried out. Of particular interest would be ability to intervene in vivo through a light guide in one specific circuit (increasing or decreasing NMDAR or other receptor function, for example) while a behavior, task, or particular pathology is monitored. Right now this can only be done through microinjection of pharmacological probes, which while useful, carries numerous caveats.

While this manuscript does not establish new precedent (a number of papers describing opto-switches on receptors have been published), this is certainly an intriguing application of the technology, in that the authors place the switchable side chain at key regions that control receptor function. Overall, the ability to confer spatiotemporal precision through light activation is a valuable tool for the field and could be, in time, influential, assuming the method is incorporated into approaches used by others to explore receptor function. I predict multiple labs are likely to utilize and build on these kind of approaches.

I have only one major comment that the authors should consider.

1) Isomerization of azobenzene in place of GluN1-Y647 alters single channel currents, which is interpreted to reflect a change in channel conductance. There is insufficient analysis to conclude this, as it looks as though the openings have become so brief and flickery that they no longer have time to reach full amplitude. There are several methods for determining this, such as the Patlak mean-low variance method, to find clear openings long enough to trust the measured amplitude. The all points histogram cannot easily be interpreted, and the channels are not shown at sufficient resolution to make a clear case for a change in conductance. I would recommend more careful analysis and display of these channels, and discussing the possibility that the apparent decrease in conductance reflects brief events that do not have sufficient time to reach full amplitude.

The same caveat applied to the fifth paragraph of the Discussion, which mentions that "[…]the dual increase in channel P_o_ and conductance[…]" Overall, the analyses of these single channel data could be better developed, and results presented with more caution and care.

*Reviewer #3:*

The manuscript by Klippenstein V. et al., describes the photocontrol of NMDA receptors by photoswitchable unnatural amino acids (UAA). This work relies on the group's extensive experience with NMDA receptors and, more recently, the incorporation of UAA within the receptors. They describe a unique method by which they create several light-gated receptors (inactivated and potentiated). This method bypasses several limitations of other techniques. For example, UAAs can be incorporated in many regions of the subunit, in particular transmembrane domains that are typically inaccessible by other soluble photoswitches. Secondly, this method ensures a homogenous population of light-ready receptors at the membrane, as opposed to other methods that rely on random binding of the photoswitch to receptors at the membrane (yielding a mixed population). I think this is a very powerful technique and a highly valuable addition to the growing palette of NMDA-related optical tools.

Could the authors show the -trans and -cis absorption spectra? Though, they cite Hoppmann 2011/14 as the source for PSAA, I could not find these data in the papers. This is important for understanding the effect of blue (460 nm) vs. green (520 nm) light effects shown in Figure 3; and for the reason the authors have used blue and uv light as the main sources for photoswitching, rather than uv and green, as is typically done with azobenezene photoswitches. The authors show a much higher off-switching rate when blue light was used in comparison to green light. In addition they show that whereas blue light yields full recovery from the light-induced block, green light does not. They attribute this to the fact that green light might induce a mixture of -trans and -cis populations. By examining the absorption spectra of related azobenzene compounds it is evident that green light is much less absorbed by -trans, but readily absorbed by -cis. This illumination then yields a more homogenous -trans population. However, blue light is absorbed by both isomers, which should have given then a higher mixture of -trans and -cis isomers. Could the authors explain this observation?

The presence of a single homogenous receptor population at the membrane is a unique strength of this method compared to others. However, it appears that this method yields very low expression of the receptors: the average currents for PSAA-treated cells expressing GluN1-P532Amber is ~5 times smaller than the average current obtained from wt receptors (Figure 2—figure supplement 2 460 pA vs. 2200 pA, respectively). This is also observed for the Y647 mutant (Figure 6—figure supplement 2) and, as noted by authors, most severe in the case Y535 (though no data shown). Could the authors aggregate all currents (raw I, pA) for all mutants and wt in a single panel? From a user point of view, this might help in choosing which mutant to proceed with based on its expression levels in HEK cells.

I do not understand the authors reasoning behind the PAM binding experiments. They have mutated the receptor's Proline at position 532 and claim that since PAM is no longer able to potentiate the current, then PSAA must be there. I would argue that the photo-effect (inactivation in the case of 532) is sufficient to claim that the PSAA is there, otherwise the channels don't express at the membrane (as they show) and light would have no effect on the current (as they show for wt receptors). Could it be that the simple removal of Proline (say by any other a.a.) would completely abolish the effect of PAM? If so, then I would argue that this is a likely reason why PAM no longer works on this mutant channel.

Figure 2—figure supplement 3, panel E is confusing (likely erroneous). The authors show that under all conditions (w/o PAM at different illuminations) the channel is inhibited. Isn't the inactivation removed upon blue light illumination? Could this be re-examined or explained?

---

## [Author Response]

We thank the three reviewers and the editor for their critical reading and comments on our manuscript. We were particularly pleased to learn that both the quality of the work and the writing were praised. We also appreciate the suggestion of moving the article into the Tools and Resources section. According to these comments, and as detailed below, we have made substantial revision on our manuscript. Specifically, the following major changes were made:

1) As required, we put larger emphasis on the presented methodology itself, its advantages, technical requirements and biological applications. For that purpose, we have greatly extended the Discussion along the three following lines (and the end of the revised Introduction): i) how PSAA mutagenesis differs from classical site-directed mutagenesis; ii) what sorts of questions could be answered using photoswitchable NMDARs (and neuronal receptors in general); iii) what it exactly needed to apply the PSAA technology in more native systems. Writing these sections has been truly instructive and gratifying for us, in particular by allowing us to elaborate on the importance of the methodology and its future applications. We think this should be informative and appealing for the general reader. It also fits well with the new affiliation of our manuscript to the Tools and Resources section of eLife.

2) We have strengthened the section on light-induced changes in receptor channel permeation (i.e. changes in single-channel conductance) by adding new experimental data (noise analysis; see subsection “Optical control of gating and permeation properties with PSAA within the TMD”, second paragraph and new Figure 7—figure supplement 2).

3) We have included more experimental information about the optochemical properties of PSAA and the expression levels following its introduction within different sites of the receptor (new Figure 2—figure supplement 1). In particular, as requested, we now provide *trans-* and *cis-* absorption spectra of the PSAA photoswitch (new Figure 3—figure supplement 1).

4) We have toned down certain interpretations or comments to better match the data (single-channel recordings; speed of light-induced receptor activity changes).

*Reviewer #1:*

*[…] However, beyond demonstration of proof of principle, the paper does not greatly advance our understanding of receptor function. Two of the sites, in the ligand binding domain dimer interface, and in the amino terminal domains were previously demonstrated to be important regulators of receptor activity, and UAA photo switching yields no new information about receptor function. The third site, in a hydrophobic cluster near the gate of the ion channel domain, yields receptors with light gated changes in single channel open probability, and very subtle changes in channel block which the authors over interpret as a change in permeation properties, consistent with light triggered structural changes in the ion channel domain, but again the results give no new insight into receptor function or the underlying structural changes; in a sense the approach is not much of an advance over traditional mutagenesis in that it does not yield structurally informative insights into channel function. An additional caveat is that the size and chemistry of the labeled residue introduced by UAA incorporation is hugely different from that of the wild type side chain, and thus in either of the light triggered conformations, receptor function is non-native.*

*A final caveat is that the UAA azobenzene approach will likely be difficult, perhaps impossible, to use in vivo and if this is true, the elegant recordings from transiently transfected HEK cells will have little impact beyond biophysical studies on recombinant receptors. Much of the excitement in optogenetics comes from the ability to alter receptor function in intact tissues, and in behaving animals. This is a beautiful paper, but the approach is a niche study which I think will not have great impact on the field.*

Our article is now qualified as a Tools and Resources article and, accordingly, in our revision we put stronger emphasis on the methodological aspects of the PSAA approach. In particular, as described in the general comments above, we have extensively modified the Discussion to include sections on the pros and cons of the methodology, its utility to address key biological questions and its challenges for in vivo applications. By providing the first demonstration of real-time detection of molecular rearrangements due to reversible light-switching of a single-amino acid side chain inserted into a neuronal receptor, we believe that our work provides an important and significant advance in the fast evolving field of optopharmacology, which is still in need for novel approaches allowing the manipulation of specific receptor subtypes in a direct manner and with high temporal resolution. Although the focus of our study is set on the presentation and characterization of the PSAA methodology itself, we think it also provides novel and interesting insights into the structural mechanism of NMDAR function: i) by identifying a single position (P532) in the GluN1 agonist-binding domain hinge region that allows specific photo-tuning of glycine co-agonist dissociation kinetics (see Figure 4); and ii) by identifying a conserved transmembrane hydrophobic cavity in the ‘upper’ part of the channel as a critical element controlling the energetics of both channel gating and permeation (see Figure 6 and Figure 7). It is no big surprise that this region – facing opposite to the SYTANLAAF M3 channel gate – influences channel open-closed transitions. That this channel region also influences channel permeation is more unexpected and indicates that the M2 P-loop region may not be the sole important contributor to ion permeation through glutamate receptor ion channels. In that respect, we consider that our data, including the newly acquired noise analysis results, strongly support our conclusion that PSAA introduction allows to photomodulate permeation properties. As discussed in the revised manuscript, our observed changes in Mg^2+^ block IC_50_, of modest amplitude, yet significant, are strong indicators of altered interactions between Mg^2+^ ions and the channel lumen. Similarly, our single-channel conductance analysis points to large (>2-fold) changes in unitary conductance upon photoisomerization of GluN1-Y647PSAA (see new Figure 7—figure supplement 2). These results demonstrate the first use of single photo-active amino acids to directly control transmembrane domain motions and ion channel properties. In conclusion, we developed and implemented a new methodology, and show its utility to probe the biophysics of a key neurotransmitter receptor.

Finally, we are grateful to the reviewer for pointing out the challenges of the in vivo use of the UAA methodology. Implementing the UAA approach in native preparations is indeed challenging, yet not impossible. As a matter of fact, many groups are currently working on this issue and several recent reports have provided proof-of-principle feasibility for the UAA approach in vivo (including in a whole living animal and targeting a brain protein). The generation of transgenic mice with a heritable expanded genetic code – i.e. mice directly incorporating the necessary tRNA/synthetase genes for Amber stop codon suppression in their genome – is certainly one of the most powerful developments that will greatly facilitate in vivo UAA applications. Mouse strains allowing incorporation of the photocrosslinker AzF (Chen et al., 2017, Cell Res) or the post-translationally modified lysine N^ε^-acetyl-lysine (AcK; Han et al., 2017, Nat Comm) have just become available. We expect a mouse strain for PSAA incorporation to become available in the near future. We (and many others) are much excited about these recent developments for future biological applications to study brain function using UAAs, including PSAAs. In the revised manuscript, we describe and discuss in detail these aspects related to the in vivo implementation and application of the UAA/PSAA approach (see the revised Discussion, last paragraph) (see also our general comment at the beginning of this rebuttal letter).

*Reviewer #2:*

*[…] I have only one major comment that the authors should consider.*

*1) Isomerization of azobenzene in place of GluN1-Y647 alters single channel currents, which is interpreted to reflect a change in channel conductance. There is insufficient analysis to conclude this, as it looks as though the openings have become so brief and flickery that they no longer have time to reach full amplitude. There are several methods for determining this, such as the Patlak mean-low variance method, to find clear openings long enough to trust the measured amplitude. The all points histogram cannot easily be interpreted, and the channels are not shown at sufficient resolution to make a clear case for a change in conductance. I would recommend more careful analysis and display of these channels, and discussing the possibility that the apparent decrease in conductance reflects brief events that do not have sufficient time to reach full amplitude.*

The GluN1-Y647 single-channel openings (in the dark or blue illumination) are indeed very brief and difficult to analyze, and we acknowledge that our experimental conditions (including filtering) likely limit the resolution to make clear statements about unitary conductance. We have reconsidered different methodologies to (re)analyze our data and, eventually, decided to perform a whole-cell noise analysis to get estimates of the apparent single-channel conductance. To do so, additional whole-cell recordings at a higher sampling rate (20 kHz) and lower filtering (10 kHz) for the two TMD mutants GluN1-F654PSAA and GluN1-Y647PSAA were performed. We have included the results of these experiments into a new supplementary figure (new Figure 7—figure supplement 2).

Overall, our new data as obtained from the macroscopic current noise under dark or UV conditions, provide further support for (i) a change in single-channel conductance for the GluN1-Y647PSAA mutant comparing the dark (PSAA in the *trans*-isoform) and the UV state (PSAA in the *cis*-isoform); and (ii) the absence of this effect for the GluN1-F654PSAA mutant. As hinted by close inspection of the single-channel recordings, the constrained ion channel state resulted in a 50% lower conductance level (~27 pS) for the Y647PSAA mutant in the dark compared to exposing the same cell to UV light, which restored the single-channel conductance into a more WT-like level (~57 pS). This 2-fold increase of single-channel conductance stands in contrary to GluN1-F654PSAA receptors with no eminent changes in conductance levels when switching between the two photoisomers, a result well aligned with our single-channel recordings and all-point amplitude histogram analysis (see Figure 7 and Figure 7—figure supplement 1). We believe that these experiments are a good addition to the effects we have observed in our single-channel data and that, altogether, they support our conclusion that PSAA allows for light-induced changes in receptor channel permeation. These results contribute to a clearer understanding of the events occurring during the different photoisomerization states on the level of the TMD ion channel, which strengthens our revised version of the manuscript.

As a final note in this point, the way we have analyzed the whole-cell noise is described in a detailed way in the Materials and methods section of the revised manuscript:

“For the analysis of whole-cell noise, currents were sampled at 20 kHz and low-pass filtered at 10 kHz. […] This ratio was determined for each individual cell. Relative changes in γ_noise_ (Rγ) between UV and dark were then calculated, for each individual cell, using the formula: Rγ = γ_noise_ (UV)/γ_noise_ (dark).”

*Reviewer #3:*

*[…] Could the authors show the -trans and -cis absorption spectra? Though, they cite Hoppmann 2011/14 as the source for PSAA, I could not find these data in the papers. This is important for understanding the effect of blue (460 nm) vs. green (520 nm) light effects shown in Figure 3; and for the reason the authors have used blue and uv light as the main sources for photoswitching, rather than uv and green, as is typically done with azobenezene photoswitches. The authors show a much higher off-switching rate when blue light was used in comparison to green light. In addition they show that whereas blue light yields full recovery from the light-induced block, green light does not. They attribute this to the fact that green light might induce a mixture of -trans and -cis populations. By examining the absorption spectra of related azobenzene compounds it is evident that green light is much less absorbed by -trans, but readily absorbed by -cis. This illumination then yields a more homogenous -trans population. However, blue light is absorbed by both isomers, which should have given then a higher mixture of -trans and -cis isomers. Could the authors explain this observation?*

We were referring to the spectra published in Hoppmann et al. (ChemBioChem, 2011; Figures S8 and S10, where PSAA was introduced into a model peptide). It is true that most studies use UV and green light as the main sources of azobenzene photoswitching, although UV/blue is not unprecedented (see, for instance, Browne et al., PNAS, 2014). The reason for why in the present study we have used UV/blue is purely empirical, meaning that we found UV/blue to be better than UV/green to achieve full and ‘fast’ reversibility. We now provide experimental evidence to why UV/blue may be more appropriate than UV/green to photomanipulate PSAA-containing proteins. Indeed, as requested, we acquired the *trans*- and *cis*- absorption spectra of PSAA. These new experiments were performed on free PSAA diluted either in our extracellular (Ringer) solution to be as close as possible to the recording conditions or in isopropanol (Ye et al., Nat Chem Biol, 2009) to mimic a hydrophobic environment (since PSAA is buried in the protein). The results are shown below. From these spectra, it can be deduced that i) the absorption peak of the *trans*-form is around 330 nm but at this wavelength the trans form also absorbs significantly; ii) 360 nm, the wavelength used to photoisomerize PSAA in the *cis*-configuration, appears appropriate since it is a range where the absorbance ratio is the most favorable for the *trans*-form; iii) blue light is as good (Ringer), or even better (isopropanol), than green light to flip back *cis*-PSAA to the *trans*-form. Obviously, the type of environment encountered by PSAA significantly influences its photochemical properties. In particular, when introduced at a site poorly accessible to solvent (at interfaces or TMD locations, as in the current study), blue light should be preferred to green light for photoconversion from *cis*- to *trans*-. These newly-acquired results have been included in the revised manuscript as a new Figure (Figure 3—figure supplement 1), thus providing better completeness of the PSAA photochemical characterization,

*The presence of a single homogenous receptor population at the membrane is a unique strength of this method compared to others. However, it appears that this method yields very low expression of the receptors: the average currents for PSAA-treated cells expressing GluN1-P532Amber is ~5 times smaller than the average current obtained from wt receptors (Figure 2—figure supplement 2 460 pA vs. 2200 pA, respectively). This is also observed for the Y647 mutant (Figure 6—figure supplement 2) and, as noted by authors, most severe in the case Y535 (though no data shown). Could the authors aggregate all currents (raw I, pA) for all mutants and wt in a single panel? From a user point of view, this might help in choosing which mutant to proceed with based on its expression levels in HEK cells.*

We thank the reviewer for pointing out the missing information about the overall expression levels of all mutants tested. We have included this information into a scatter dot plot (new Figure 2—figure supplement 1) representing the maximal peak currents of WT receptors and all Amber mutants that have shown any kind of photomodulation. For the sake of completeness, we also provide the information about the expression level of the GluN1-Y535 mutant, which is mentioned in the manuscript.

*I do not understand the authors reasoning behind the PAM binding experiments. They have mutated the receptor's Proline at position 532 and claim that since PAM is no longer able to potentiate the current, then PSAA must be there. I would argue that the photo-effect (inactivation in the case of 532) is sufficient to claim that the PSAA is there, otherwise the channels don't express at the membrane (as they show) and light would have no effect on the current (as they show for wt receptors). Could it be that the simple removal of Proline (say by any other a.a.) would completely abolish the effect of PAM? If so, then I would argue that this is a likely reason why PAM no longer works on this mutant channel.*

Yes exactly, the aim was to show that PSAA incorporation eliminates PAM binding and action. In order to confirm the presence of the PSAA at the GluN1-P532 site, we have analyzed two different effects: i) The potentiation by PAM; and ii) the photomodulation properties induced by PSAA. A proline residue at this position (WT condition) induces a strong PAM-potentiation with no detectable photoinactivation. The presence of the PSAA, however, abolishes the PAM effect entirely (probably due to the disruption of the PAM binding site) and critically, still retains the photoinactivation properties. This result strongly confirms the presence of the PSAA at this Amber codon position of interest, since the photoinactivation effect can be only due to the presence of a photoactivatable chemical group (the azobenzene moiety). To further verify our hypothesis, we have performed an additional experiment that allows the introduction of a Tyrosine residue at the very same position. To do so, we have co-transfected a different set of tRNA/synthetase pair that drives the suppression of Amber stop codons by Tyrosine (as described in Ye et al., ChemBioChem, 2013). Tyrosine, such as PSAA, is an aromatic amino acid. Its introduction at the GluN1-P532 position likewise abolished the potentiation by PAM (see Figure 9), certainly due to a disrupted PAM binding site. This effect strongly supports the hypothesis that aromatic groups at this ABD intra-dimer hinge abolish the typical PAM effect of current potentiation. Taken these results together, we believe that the dual effect of the missing PAM potentiation and the persisting photoinactivation is a convincing readout of PSAA incorporation at the GluN1-P532 site. Figure 9 has not been included in the revised manuscript, but we have modified the legend of the existing PAM figure for enhanced clarity.

Author response image 1.Tyrosine at the GluN1-P532 Amber site disrupts the binding site of PAM.(**a**) Example trace demonstrating that PAM does not potentiate mutant receptors with tyrosine at the GluN1-P532 site (co-expressed with GluN2A-WT subunits). Light at 365 or 460 nm does not induce any changes of receptor activity. (**b**) Mean fold-potentiation by PAM for GluN1-P532Y receptors (1.01 ± 0.03, n=3).**DOI:**
http://dx.doi.org/10.7554/eLife.25808.026

*Figure 2—figure supplement 3, panel E is confusing (likely erroneous). The authors show that under all conditions (w/o PAM at different illuminations) the channel is inhibited. Isn't the inactivation removed upon blue light illumination? Could this be re-examined or explained?*

We apologize for the confusion created by this figure. The confusing color coding arose from the trial to distinguish between the different light / PAM protocols that were run on the WT and mutant receptors. In panel b, PAM was applied after switching the PSAA into the *trans*-state (blue light), while in panel c, the receptors were exposed to PAM following induction of the *cis*-isomer (UV light). To provide more clearness, we have now pooled the photoinactivation degrees, independently of the photostationary state of the PSAA prior PAM exposure, and removed the color coding. The same has been done for the potentiation effect of PAM as shown in panel d. The overall message of this figure is unchanged, but clarity has been improved.